# The acceleration of sea-level rise along the coast of the Netherlands started in the 1960s

Iris Keizer[1], Dewi Le Bars[1], Cees de Valk[1], André Jüling[1], Roderik van de Wal[2,3], and Sybren Drijfhout[1,2]

[1]Royal Netherlands Meteorological Institute (KNMI), De Bilt, The Netherlands
[2]Institute for Marine and Atmospheric Research Utrecht, Utrecht University, Utrecht, The Netherlands
[3]Department of Physical Geography, Utrecht University, Utrecht, The Netherlands

**Correspondence:** Iris Keizer (iris.keizer@knmi.nl)

**Abstract.**

The global acceleration of sea-level rise (SLR) during the 20th century is now established. On the local scale, this is harder to establish as several drivers of SLR play a role which can mask the acceleration. Here, we study the rate of SLR along the coast of the Netherlands from the average of six tide gauge records covering the period 1890–2021. To isolate the effects of the wind field variations and the nodal tide from the local sea-level trend, we use four generalised additive models (GAMs) which include different predictive variables. From the sea-level trend estimates, we obtain the continuous evolution of the rate of SLR and its uncertainty over the observational period. The standard error in the estimation of the rate of SLR reduces when we account for nodal tide effects and reduces further when we also account for the wind effects meaning these provide better estimates of the rate of SLR. A part of the long-term SLR is due to wind forcing related to a strengthening and northward shift of the jet stream, but this SLR contribution decelerated over the observational period. Additionally, we detect wind-forced sea-level variability on multidecadal time scales with an amplitude of around 1 cm. Using a coherence analysis, we identify both the North Atlantic Oscillation and the Atlantic Multidecadal Variability as its drivers. Crucially, accounting for the nodal tide and wind effects changes the estimated rate of SLR, unmasking a SLR acceleration that started in the 1960s. Our best-fitting GAM, which accounts for nodal and wind effects, yields a rate of SLR of about $1.7\,^{2.2}_{1.3}$ mm/yr in 1900–1919 and $1.5\,^{1.9}_{1.2}$ mm/yr in 1940–1959 compared to $2.9\,^{3.5}_{2.4}$ mm/yr in 2000–2019 (where the lower and upper bounds denote the 5th and 95th percentile). If we discount for the nodal tide, wind and fluctuation effects and assume a constant rate of SLR, then the probability (p-value) of finding a rate difference between 1940–1959 and 2000–2019 of at least our estimate is smaller than 1%. Consistent with global observations and the expectations based on the physics of global warming, our results show unequivocally that SLR along the Dutch coast has accelerated since the 1960s.

## 1 Introduction

Understanding the current and past rates of sea-level rise (SLR) is essential to make reliable sea-level projections and to adapt accordingly. In the Netherlands, the current rate of SLR is used to estimate the volume of sand that must be supplied to maintain the coastline and avoid a retreat of dunes. It also estimates how much salt and gas mining can be allowed under the Wadden

Sea. In addition, local sea-level measurements are important to evaluate sea-level projections (Vries et al., 2014). The rate of SLR could be used as an early warning indicator for adaptation measures to uncertain climate change (Haasnoot et al., 2018).

There is now *high confidence* in an acceleration of global SLR in the 20th century compared to the previous three millennia and in the period 2006–2018 compared to 1971–2018 (Fox-Kemper et al., 2021). Dangendorf et al. (2019) found the global rate of SLR to accelerate from the 1960s. More recently, Walker et al. (2022) estimated that the rates of SLR emerged from the background variability of the Common Era (0–2000 CE) in the middle of the 19th century for the globe and in the middle of the 20th century for the North-East Atlantic.

Focusing on sea-level change along the coast of the Netherlands, the existence of an acceleration of SLR is still debated (Baart et al., 2011; Wahl et al., 2013; Steffelbauer et al., 2022). There are multiple lines of evidence that an acceleration should already be detectable or will be detectable soon. The increasing thermal expansion of oceans and faster-melting glaciers and ice sheets drive the global acceleration of SLR. These mechanisms are also expected to contribute to SLR in the North Sea. However, the contribution of mass loss from the Greenland Ice Sheet is much smaller than the globally averaged contribution, due to gravitational effects (Slangen et al., 2012). The contribution of glaciers to SLR in the North Sea is below the global mean value as the North Sea is relatively close to glaciers which are mostly based on the Northern Hemisphere (Slangen et al., 2012). Additionally, the ocean dynamic sea level is expected to rise along the North-East Atlantic (Lyu et al., 2020; Hermans et al., 2022) and dynamic sea-level projections based on climate models from the Coupled Model Intercomparison Project (CMIP5 and CMIP6) also expect an acceleration. Combined, the expectation for the climate-driven sea-level change along the Dutch coast is close to the global mean changes (Vries et al., 2014; Fox-Kemper et al., 2021).

The data availability along the Dutch coast is much better than for reconstructed global sea level (Dangendorf et al., 2019; Frederikse et al., 2020). There are six tide gauges, homogeneously distributed along the coast, measuring sea level with very little missing data since at least 1890 which is favourable for a study of regional SLR acceleration. We study the average of the six tide gauges to estimate the rate of SLR along the Dutch coast. Averaging helps to increase the signal-to-noise ratio and avoids considering processes that drive differences for local rates of SLR like vertical land motion and small-scale ocean processes. Furthermore, because of their proximity, long-term changes at these stations are expected to be similar (e.g. the differences are not resolved in the CMIP6 climate models), and an average for the Dutch coast is sufficient to weigh up adaptation choices. However, the rates of SLR for the individual tide gauges are included in App. B. The issue with detecting a regional acceleration of SLR comes from the large interannual to multidecadal variability from atmospheric forcing, especially important for shallow seas like the North Sea (Gill, 1982; Hermans et al., 2020) and from similar variations in local steric sea level (Bingham and Hughes, 2012). Detecting the acceleration of SLR requires understanding the sources of interannual-to-multidecadal variability and removing them from tide gauge records (Haigh et al., 2014). To this end, various authors have used multilinear regression models between sea levels and atmospheric variables like sea surface pressure gradients, zonal and meridional surface wind velocities and, at times, precipitation. For example, this approach was applied to Cuxhaven in the German Bight by Dangendorf et al. (2013) and multiple regions by Calafat and Chambers (2013). Nevertheless, there is no generally agreed approach for detecting a SLR acceleration from tide gauge stations. Sometimes the observed records are extended by sea-level projections and the acceleration is defined as a rate of SLR significantly larger than observed, which

only allows for finding an acceleration in the future (Haigh et al., 2014; Dangendorf et al., 2014a). Some studies compared the rates of linear SLR over two different periods (Calafat and Chambers, 2013; Steffelbauer et al., 2022) and others fitted a second-degree polynomial to the data (Haigh et al., 2014; Dangendorf et al., 2019). In general, the sea-level variability due to atmospheric forcing is estimated first by linearly detrending the time series. After that, the variability is removed from the sea-level data before estimating the trend and acceleration. Many previous studies of SLR in the North Sea did not find evidence of a significant SLR acceleration (Calafat and Chambers, 2013; Wahl et al., 2013; Haigh et al., 2014; Ezer et al., 2016) whereas Steffelbauer et al. (2022) did. To detect the acceleration of SLR in the North Sea, Steffelbauer et al. (2022) analysed the 100-year time series (1919–2018) of eight tide gauges and found a common breakpoint in the early 1990s. The average rate of SLR of the stations increases at the breakpoint from $1.7 \pm 0.3$ to $2.7 \pm 0.4$ mm/yr, which implies an acceleration of SLR. However, the prior distribution adopted for the rate of SLR before and after the breakpoint assumes that the latter rate can not be smaller than the former rate, which implies that acceleration is assumed from the beginning.

In this paper, we use a new time series approach which uses a Generalised Additive Model (GAM), which allows us to estimate a nonlinear trend and the optimal multilinear regression model simultaneously. The nodal tide and zonal and meridional wind are included in the GAM as predictive variables. Both the zonal and meridional wind are used to reduce the uncertainty in the estimated rate of SLR. Other authors did not always include the nodal tide as a predictive variable. Using the GAM, we avoid making strong assumptions about the shape of the sea-level trend like the piecewise linear shape assumed by Calafat and Chambers (2013) and Steffelbauer et al. (2022). The sea-level trend is obtained as a smooth curve representing the long-term change in the data. This curve is differentiated to compute the rate of SLR as it evolved over the observational period; this has not been obtained before. We also apply a rigorous parametric bootstrap method to estimate the uncertainty in the rate of SLR, which avoids the assumption that the noise is serially uncorrelated. Furthermore, comparing estimates of the rate of SLR with and without the effects of wind and nodal tide allows us to study the influence of these processes on SLR. We also discuss the physical mechanisms driving the wind-driven sea-level variations in the North Sea.

## 2 Data

### 2.1 Tide Gauge Observations

Annual-mean sea-level measurements are used as the average of the six reference tide gauges along the coast of the Netherlands: Delfzijl, Den Helder, Harlingen, IJmuiden, Hoek van Holland and Vlissingen (Fig. 1a). These stations are used for operational sea level monitoring because of their extended temporal coverage and homogeneous distribution along the Dutch coast (Baart et al., 2019). The measurements are made by Rijkswaterstaat and provided by the Permanent Service for Mean Sea Level and set to the Revised Local Reference (Holgate et al., 2013). They were retrieved on November 1st, 2021 from http://www.psmsl.org/data/obtaining/. The readings at these stations start between 1862 and 1872 and are gauged with respect to the mean sea level. However, the data before 1885 are gauged with respect to readings of the mean tide which could result in a jump in the data (Woodworth, 2017). Therefore, we only use the tide gauge data after 1890 as was done for Frederikse and Gerkema (2018); Baart et al. (2019).

## 2.2 Atmospheric Reanalysis

We use the monthly mean zonal and meridional wind at 10 m and atmospheric pressure at sea level from two atmospheric reanalysis products. The first product, the ERA5 reanalysis, from the Copernicus Climate Change service Climate Data Store, is available from 1979 to 2022 with a backward extension to 1950 (Hersbach et al., 2020; Bell et al., 2021). ERA5 has a spatial resolution of 0.25°×0.25°. The second product, the Twentieth Century Reanalysis Version 3 (20CRv3) from the National Oceanographic and Atmospheric Administration (NOAA), is available from 1836 to 2015 (Slivinski et al., 2019). The data from this analysis has a spatial resolution of 1.0°×1.0°.

## 3 Method

### 3.1 Statistical Models

Four statistical models were developed and used to separate the influence of different chosen predictive factors on SLR and to extract the resulting background sea-level trend. All models are based on the Generalised Additive Model (GAM, Hastie and Tibshirani (2017); Wood (2020)) and are estimated by penalised maximum likelihood. Compared to a multi-linear regression model, a GAM replaces the strict assumption of a linear or quadratic shape of the sea-level trend by a sum of many smooth functions. This offers the advantage that we are not required to make a priori assumptions about the shape of the sea-level trend. In our four models, the GAM represents the annual-mean sea level averaged over the six tide gauges as a smooth curve (a linear combination of many smooth cubic B-spline basis functions) plus terms representing the influence of the predictive variables. An overview of the four models and their mathematical description is given in Tbl. 1. The smooth curve (trend), given by the first term in the equations in Tbl. 1, represents the background variation in sea level to be estimated; its exact meaning depends on the choice of the predictive variables. Its smoothness is controlled by a penalty term subtracted from the log-likelihood, which is proportional to the time-integral of the squared curvature of the smooth term (Wood, 2020). The penalty term was assigned a weight tuned to match the variance of the smooth curve to the variance of a 30-year average.

The first model (*Tr*) estimates the sea-level trend only without using any predictive variables. This setup makes no assumptions about the drivers of SLR. We use this model as a reference to evaluate the improvements achieved by increasing the model complexity. In the second model, the influence of the lunar nodal tide on sea level is added (*TrNt*). A sinusoidal wave with unknown amplitude and phase and a fixed period of 18.613 years, the period of the nodal tide potential, are included as a predictive variable for the nodal tide in the GAM. There has been some debate in the literature about the best way to estimate the influence of the nodal tide on the sea level in the North Sea. Using linear regression to estimate the effect of the nodal tide along the Dutch coast shows an increased magnitude and shift in the phase compared to the equilibrium tides (Baart et al., 2011). However, using a closed sea-level budget, Frederikse et al. (2016) suggested there is no indication that the nodal tide deviates from the equilibrium tide in the North Sea between 1958 and 2014. We find that assuming equilibrium tides leaves a large amount of energy in the spectrum close to the period of the nodal tide (see App. A). Therefore, we decide to use a linear regression model with an undetermined phase and amplitude but a fixed period as in Baart et al. (2011) even though it might

remove some additional variability around the period of nodal tides. Using this second model, the influence of the nodal tide on the trend and variability of sea level can be studied.

The third and fourth models combine trend, nodal tide and wind effects. For the third model (*TrNtW*), wind effects are included by adding $u|u|$ and $v|v|$ (Tbl. 1), where $u$ and $v$ are, respectively, the zonal and meridional wind from reanalysis obtained from the closest grid cell of each tide gauge and averaged for the six stations (Fig. 1a). The wind expression is inspired by the wind stress formulation (Dangendorf et al., 2019). Along the Dutch coast, the zonal wind is much more important for sea-level changes than the meridional wind (Figs. 7 and 8 from Frederikse and Gerkema (2018) and Fig. 4 from Dangendorf et al. (2014a). However, including both the zonal and meridional wind components reduces the uncertainty in the estimated rate of SLR more than only including the zonal component. The fourth model (*TrNtPd*) uses a large-scale pressure gradient as the predictive variable for the wind effect on sea level. As in Dangendorf et al. (2014b), we compute the Pearson correlation coefficient between linearly detrended sea level along the Dutch coast and atmospheric pressure at sea level (Fig. 1b). This shows a similar pattern as was previously obtained for the German bight (Dangendorf et al., 2014b). The pattern is characterised by a region of negative correlation over Scandinavia and a positive correlation over southern Europe/northern Africa. Each of these regions defines a box where the average pressure is computed. Then, instead of using the pressure in both boxes as predictive variables as in Dangendorf et al. (2014b), we take the difference between the southern and northern boxes. This adds only one variable to the model and is physically motivated by the fact that the pressure gradient is to some extent related to wind by geostrophy. We combine the variables representing wind effects from the two reanalysis datasets using a linear bias correction method (Casanueva et al., 2013) to obtain one dataset covering the full period of atmospheric data from 1836 to 2022. The ERA5 dataset is used as reference data for the correction. The mean and standard deviation of the 20CRv3 pressure and wind time series are adjusted to match the means and standard deviations of 20CRv3 and ERA5 over the overlap period 1950–2015.

## 3.2 Analysis of Model Output

Using our four GAMs including different predictive variables enables us to study the background sea-level trend, the influence of the nodal tide on sea level and the wind influence on sea level. The wind influence on sea level can be obtained from the results of *TrNtW* and *TrNtPd*. It is described by the third plus fourth term (*TrNtW*) or the fifth term (*TrNtPd*) in the model equations given in Tbl. 1. We obtain the regression coefficients from our GAMs over the period from 1890 to 2021. Using these coefficients and the wind data, we can obtain the wind influence on sea level from 1836 to 2022, the period covered by the atmospheric reanalyses. From this, we obtain the trend of wind-driven sea level using a 3rd-degree polynomial fit to the annual-mean data. Also, a spectral analysis is performed on the detrended annual-mean data. The spectra are obtained using a multitaper method (Lees and Park, 1995). To obtain the low-frequency wind influence on sea level, the detrended annual-mean sea level data is smoothed using local polynomial regression (LOWESS, Cleveland and Devlin (1988)) with a window of 21 years that effectively removes high-frequency variability.

Using our four statistical models, we obtain the background sea-level trend (the first term in the equations in Tbl. 1. As a next step, the rate of SLR is obtained by differencing these estimated smooth sea-level trends using a three-year step. Since a

**Table 1.** Overview of the equations describing the four GAMs and summary of the statistical model performance. In the model equations, $\eta_t$ is the average sea level for the year $t$, $\phi_{jt}$ is the value of the smooth B-spline basis function $\phi_j$ for the year $t$ and $\alpha_j$ is the corresponding coefficient. $\beta_j$ are the coefficients of the predictive variables for wind effects. $A$ is the amplitude , $T = 18.613$ y is the period and $\varphi$ is the phase of the nodal tide. In these equations, the first term describes the sea level trend, the second term describes the influence of the nodal tide on sea level and the third plus fourth or fifth term gives the wind influence on sea level. The number of degrees of freedom includes the number of predictive variables and the number of basis functions used by the B-spline method. The deviance is a generalisation of the sum of squares of residuals used to compare linear regression models.

| Model | Components | | | | | | Performance | |
| | sea level | trend (spline) | nodal tide | zonal wind | meridional wind | pressure difference | Degrees of freedom | Deviance |
|---|---|---|---|---|---|---|---|---|
| *Tr* | $\eta_t =$ | $\sum_j \alpha_j \phi_{jt}$ | | | | | 4.7 | 1167.0 |
| *TrNt* | $\eta_t =$ | $\sum_j \alpha_j \phi_{jt}$ | $+A\sin(2\pi t/T+\varphi)$ | | | | 6.6 | 1033.0 |
| *TrNtW* | $\eta_t =$ | $\sum_j \alpha_j \phi_{jt}$ | $+A\sin(2\pi t/T+\varphi)$ | $+\beta_1|u_t|u_t$ | $+\beta_2|v_t|v_t$ | | 8.6 | 423.0 |
| *TrNtPd* | $\eta_t =$ | $\sum_j \alpha_j \phi_{jt}$ | $+A\sin(2\pi t/T+\varphi)$ | | | $+\beta_3\Delta p_t$ | 7.6 | 652.0 |

window of three years is used, the rates cannot be computed for the first and last years of the time series. The rates of SLR resulting from the different models do not include the same physical processes. The resulting rates of *TrNtW* and *TrNtPd* do not include the contribution from wind and nodal effects and *TrNt* does not include nodal effects while *Tr* includes all processes.

### 3.3 Uncertainty Computation

To estimate our models from the data, we use a generic method for likelihood-based estimation of GAM (Wood, 2020). It treats the unknown noise terms, the residuals, as independent identically distributed normal random variables. However, checks of the residuals reveal that they are serially correlated, so the independence assumption is not warranted. This does not invalidate the method: since only marginal parameters are estimated, the estimator is consistent under weak assumptions on the dependence (Section 2 of Cox and Reid (2004)). However, serial dependence of the noise affects the covariance of the estimated model parameters, so for deriving confidence intervals and for testing hypotheses, we must account for it. Our estimator for the rate of SLR (the derivative of the smooth spline estimate of the variation in sea level) is particularly sensitive to low-frequency components of the noise. Our error analysis must account for these subtle aspects of serial dependence. Therefore, we apply a parametric bootstrap method based on the noise spectrum, similar to the Wild Bootstrap version of the technique in Kirch and Politis (2011): we estimate the noise spectrum, using the same method as described in the previous section and generate random instances of the gaussian process having this spectrum. From these, we obtain instances of the sea level time series by adding the estimates of the non-random terms. Then we apply the GAM-based estimator for our models to each of these instances to obtain an estimate of the rate of SLR. We use 10000 iterations for the bootstrap method in order to obtain convergence. This sample of estimates is used to derive the error statistics and to test hypotheses.

However, because the estimate of the rate of SLR is sensitive to low-frequency noise, we cannot assume that the noise spectrum is sufficiently closely approximated by the spectrum of the residuals, as Kirch and Politis (2011) do. Therefore, we need to estimate the noise spectrum from the spectrum of the residuals. A simple iterative correction scheme solves this inverse problem. Given a guess of the noise spectrum, we simulate random instances of sea level time series as above. For each, we estimate the model coefficients, derive the residuals, estimate their spectra and average these estimates. Dividing this average by the guess of the noise spectrum gives the mean effect of model estimation, the quotient. The spectrum of the residuals is then corrected by dividing it by this quotient. The result is used as a guess for the next step. The iteration is initialised with the spectrum of the residuals. It converges within 3 to 5 iterations. The spectrum of the residuals and the estimated noise spectrum differ only in the low frequencies, as some of the noise in this band is absorbed in the spline term.

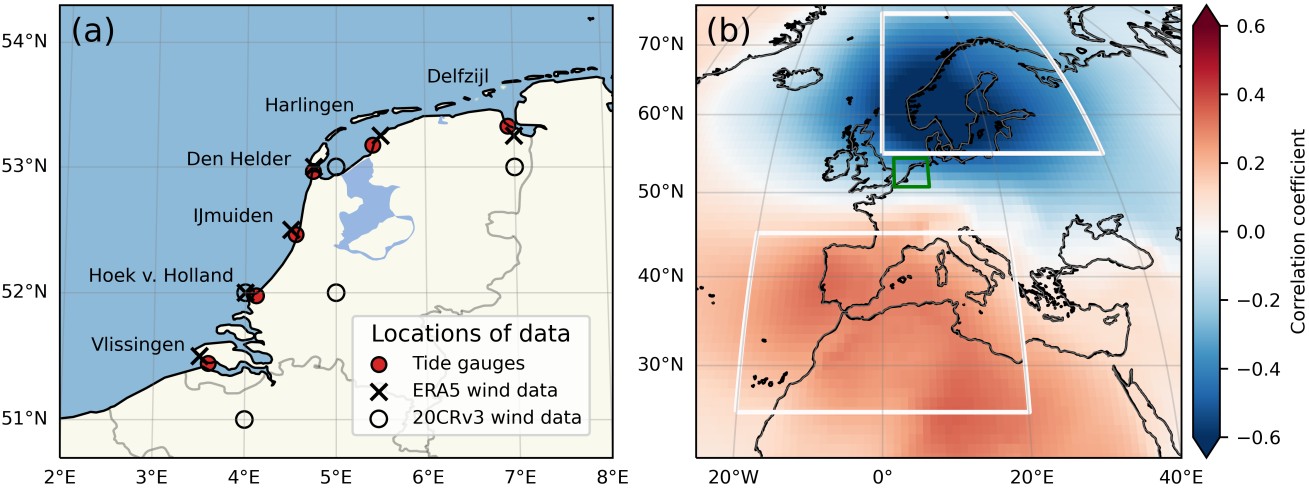

**Figure 1.** (a) Location of the six tide gauges used to define the mean sea level along the Dutch coast and of the wind data used from the atmospheric reanalyses. (b) The correlation coefficient between sea level along the Dutch coast and atmospheric pressure at sea level from 20CRv3 reanalysis between 1890 and 2015. Both variables are linearly detrended.

## 4    Results

### 4.1    Comparison of the Different GAMs

The GAM progressively better fits the data, measured by the deviance (Tbl. 1), as the complexity of the model increases (e.g., the number of predictive variables increases), measured by the number of degrees of freedom (Tbl. 1). The deviance is used to compare generalised linear models and is a generalisation of the sum of squares of residuals used to compare linear regression models (Wood, 2020). Including the nodal tide reduces the deviance by 11% and including the wind further reduces the deviance by an additional 33% for *TrNtPd* to 52% for *TrNtW* implying that the best fit is obtained for *TrNtW*. The improved

fit for *TrNtW* could be explained by the fact that here the local wind is used, whereas, for *TrNtPd*, a simplification of large-scale wind is used. The resulting fits can be seen in Fig. 2. Only the fit for *TrNtW* is shown as it strongly overlaps with the fit for *TrNtPd*. When both the zonal and meridional wind are included as predictive variables, both the deviance and the standard

error in estimating the trend are reduced compared to only including the zonal wind. Therefore, we find that using both zonal and meridional wind as predictive variables for the wind is the best choice for estimating the sea-level trend. However, when we include both northern and southern boxes as separate predictive variables (as is done by Dangendorf et al. (2014b)) instead of using their difference (as we do in *TrNtPd*) the standard error in estimating the trend is similar. Therefore, we choose to use the simplest model for estimating the sea-level trend.

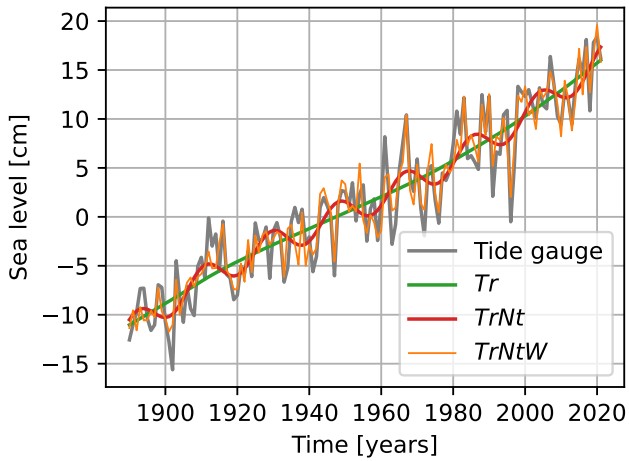

**Figure 2.** Comparison of the annual tide gauge data averaged over six tide gauges along the Dutch coast with three sea-level time series obtained from the Generalised Additive Models (*Tr*, *TrNt*, *TrNtW*). Only *TrNtW* is plotted since it overlaps strongly with *TrNtPd*, their Pearson correlation coefficient is 0.98.

**4.2    Wind Influence on Sea Level**

Figure 3a shows the resulting wind influence on sea level, where the large interannual variability stands out. From these annual-mean time series, we estimate the wind-driven sea-level trend as is shown in Fig. 3b. We find a long-term positive trend of wind influence on sea level. For the second period, 1929–2022, the wind-driven trend is 0.13 mm/yr and 0.14 mm/yr for respectively *TrNtW* and *TrNtPd*. For the first period, 1836–1928, the wind-driven trend of *TrNtW* is 0.17 mm/yr, and the trend of *TrNtPd* is

much larger, 0.42 mm/yr. An explanation for the large difference at the beginning of the time series is that the reanalysis data performance degrades further back in time due to a lack of observations. A long-term strengthening of the wind has increased the sea level along the coast of the Netherlands. This long-term strengthening of wind is consistent with the observed northward shift and increased speed of the jet stream, which could be due to a decreasing temperature gradient between the North Pole and

the equator at the height of the tropopause (Fig. 7d and 9d from Hallam et al. (2022)). The long-term influence of atmospheric drivers on SLR was studied before for the periods 1953–2003 (Fig. 2c from Dangendorf et al. (2014a)) and 1900–2011 (Fig. 12 from Dangendorf et al. (2014a)). However, we consistently find higher rates for the atmospheric driven SLR for these periods. Over the period 1953–2003, we find trends of 0.73 and 1.01 mm/yr and for the period 1900–2011 we find trends of 0.42 and 0.73 mm/yr for, respectively, *TrNtW* and *TrNtPd*. Whereas Dangendorf et al. (2014a) also find a positive trend for the period 1953–2003, the same authors find a negative trend for the period 1900–2011, contradicting our results. The differences can be due to an update in the atmospheric reanalysis (20CRv3 instead of 20CRv2).

After removing the trend from the data in Fig. 3a, a spectral analysis is performed (Fig. 3c). The spectra of the wind-impact on sea level obtained using both *TrNtW* and *TrNtPd* have a similar shape, but the total variance is larger for *TrNtW* compared to *TrNtPd* which is a result of the larger interannual variability of *TrNtW* as shown in Fig. 3a. For both methods, there is more energy in the signal for periods larger than two decades than for smaller periods. Therefore, the signals are smoothed using a local polynomial regression (LOWESS, Cleveland and Devlin (1988)) with a window of 21 years that effectively removes high-frequency variability (dashed lines in Fig. 3c). The resulting detrended and smoothed time series, Fig. 3d, show that low-frequency wind variability can raise or drop sea level by over 2 cm over a period of 2 to 5 decades. In App. C, we discuss how this low-frequency variability lags low-frequency sea-surface temperatures in the North Atlantic that have a similar pattern as the Atlantic Multidecadal Variability.

## 4.3 Rates of SLR

The rates of SLR obtained from differentiating the estimated smooth sea-level trend from each of the four models are shown in Fig. 4 and averages over different periods are shown in Tbl. 2. Reduction of uncertainty is generally the main motivation for removing variability due to known atmospheric drivers from the sea-level trend (Dangendorf et al., 2014a). The rate of change from *TrNt* has an uncertainty, averaged over time, of 0.29 mm/yr whereas *Tr* has a larger mean uncertainty of 0.45 mm/yr. Including the zonal and meridional wind (*TrNtW*) as predictive variables further decreases the average uncertainty to 0.25 mm/yr, whereas including the pressure difference (*TrNtPd*) increases the uncertainty again to 0.33 mm.yr. The standard error in estimating the trend is larger at the time series' start and end because there are fewer constraints than in the middle of the time series (Fig. 4f).

In addition to reducing the uncertainty, the wind also influences the rate of SLR itself. Both *TrNtW* and *TrNtPd* have lower rates in the first part of the 20th century compared to *Tr* and *TrNt*. From the 1960s onward, the rates of SLR of *TrNtW* and *TrNtPd* increase rapidly. The *TrNtW* model has the smallest standard error and estimates the largest rate of SLR over recent decades, which reached $2.9\,^{3.5}_{2.4}$ mm/yr over the period 2000–2019. For this model, the rate of SLR over periods before the acceleration in the 1960s is $1.7\,^{2.3}_{1.3}$ mm/yr over the period 1900–1919, $1.7\,^{2.1}_{1.4}$ mm/yr over the period 1920–1939 and $1.5\,^{1.9}_{1.2}$ mm/yr over the period 1940–1959 (Fig. 4 and Tbl. 2). We obtain the probability (the p-value) that the estimated rate difference between the periods 2000–2019 and a previous period (1900–1919, 1920–1939 and 1940–1959) would be exceeded if the sea level had changed at a constant rate (Tbl 3). For the *Tr* model, we find probabilities between 5 and 23% for the different periods. Having more predictive variables in the GAM decreases these probabilities. For the *TrNt* model, the probability is 15% when compared

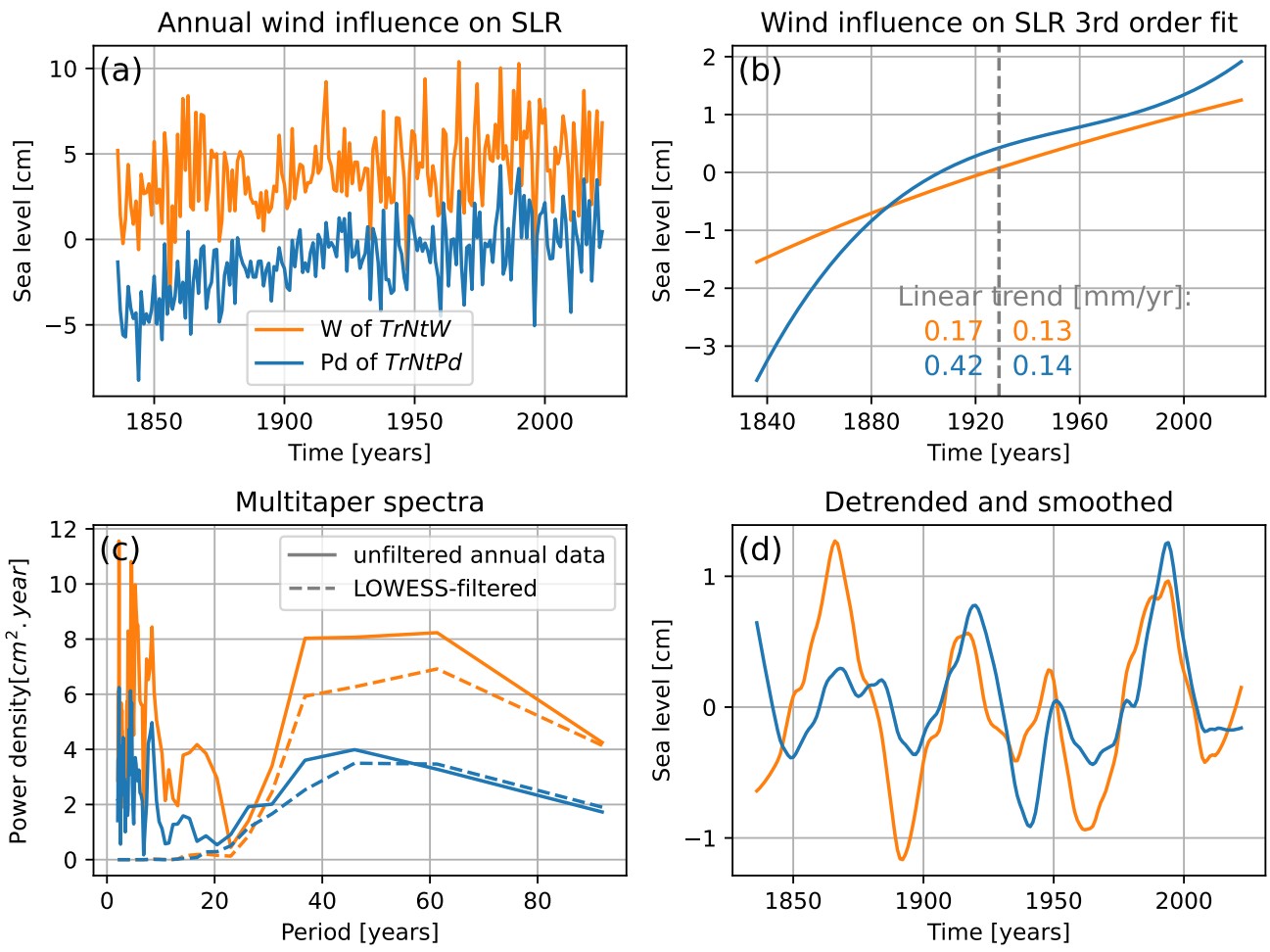

**Figure 3.** Comparison of the wind influence on sea level along the Dutch coast obtained from two different regressors: average zonal and meridional wind of the 6 tide gauge stations of Fig. 1a (*TrNtW*, orange line) and the pressure difference between the Northern and Southern boxes of Fig. 1b (*TrNtPd*, blue line) . (a) Time series of annual averages. (b) Trend computed using a 3rd-degree polynomial fit with linear trend values over the first half and the second half of the total period. (c) Spectra obtained using a multitaper method (Lees and Park, 1995). Both the detrended time series (solid lines) and the detrended and smoothed time series (dashed lines) are shown. Smoothing is obtained from a LOWESS method with a window of 21 years. (d) Detrended and smoothed time series shown in (a).

**Table 2.** The trend values are obtained by averaging the sea-level rate (Fig. 4a-d) over different periods. The lower and upper bounds are obtained by averaging the 5th and 95th percentile of the sea-level rate.

| Period | 1900–19 | 1920–39 | 1940–59 | 1960–79 | 1980–99 | 2000–19 | 1890–1959 | 1960–2021 | 1890–2021 |
|---|---|---|---|---|---|---|---|---|---|
| *Tr* | $2.2^{2.9}_{1.4}$ | $1.7^{2.3}_{1.1}$ | $1.6^{2.3}_{1.0}$ | $1.8^{2.5}_{1.2}$ | $2.3^{2.9}_{1.6}$ | $2.7^{3.7}_{1.8}$ | $1.9^{2.6}_{1.1}$ | $2.3^{3.0}_{1.5}$ | $2.1^{2.8}_{1.3}$ |
| *TrNt* | $2.3^{2.8}_{1.8}$ | $1.7^{2.1}_{1.3}$ | $1.6^{2.0}_{1.2}$ | $1.8^{2.2}_{1.4}$ | $2.3^{2.7}_{1.9}$ | $2.7^{3.4}_{2.1}$ | $1.9^{2.4}_{1.5}$ | $2.3^{2.8}_{1.8}$ | $2.1^{2.6}_{1.6}$ |
| *TrNtW* | $1.7^{2.2}_{1.3}$ | $1.7^{2.1}_{1.4}$ | $1.5^{1.9}_{1.2}$ | $1.4^{1.7}_{1.1}$ | $2.2^{2.6}_{1.9}$ | $2.9^{3.5}_{2.4}$ | $1.7^{2.1}_{1.2}$ | $2.2^{2.6}_{1.8}$ | $1.9^{2.3}_{1.5}$ |
| *TrNtPd* | $1.9^{2.4}_{1.3}$ | $1.8^{2.2}_{1.3}$ | $1.5^{2.0}_{1.1}$ | $1.5^{1.9}_{1.0}$ | $2.1^{2.6}_{1.6}$ | $2.7^{3.4}_{2.0}$ | $1.8^{2.3}_{1.2}$ | $2.1^{2.7}_{1.6}$ | $1.9^{2.5}_{1.4}$ |

**Table 3.** P-values represent the probability that the estimated rate difference between 2000–2019 and a previous period before 1960 would exceed the computed value if the actual rates were equal during these two periods. For example, for the *Tr* model, if the sea level rates were the same between 1900–1919 and 2000–2019, then there would be a probability of 23% to compute a rate difference at least as large as what we measure. On the other hand, for the model *TrNtW*, the probability of obtaining a rate difference that is at least as large as measured under the assumption that the rates are the same is smaller than 1% for all past periods considered here.

| | $r_{2000-2019}$ vs. $r_{1900-1919}$ | $r_{2000-2019}$ vs. $r_{1920-1939}$ | $r_{2000-2019}$ vs. $r_{1940-1959}$ |
|---|---|---|---|
| Statistical model | | | |
| *Tr* | 0.23 | 0.05 | 0.06 |
| *TrNt* | 0.15 | 0.01 | 0.01 |
| *TrNtW* | <0.01 | <0.01 | <0.01 |
| *TrNtPd* | 0.05 | 0.02 | 0.01 |

with the period 1900–1919 due to the higher rates of SLR of this model at the beginning of the 20th century. However, for the other periods, we find probabilities of 1%, implying that finding these rate differences would be *very unlikely* if there would have been no acceleration (Mastrandrea et al., 2011). For the *TrNtW* model, we find probabilities smaller than 1% for all periods and in the *TrNtPd* model, we find probabilities smaller than 5% and only 1% when compared with the period 1940–1959. These probabilities clearly indicate an acceleration of SLR along the coast of the Netherlands since the 1960s, which has been masked by wind-field and nodal-tide variations. This agrees with the global mean sea level that has accelerated since the 1960s (Dangendorf et al., 2019).

## 5 Discussion

By estimating the trend, nodal tide and atmospheric processes underlying the wind influence on sea level simultaneously using the GAM, we can avoid a priori assumptions about the sea-level trend, like having a linear or quadratic shape. Furthermore, the rate of SLR can be computed as a time-evolving variable over the whole observational period contrary to being calculated as

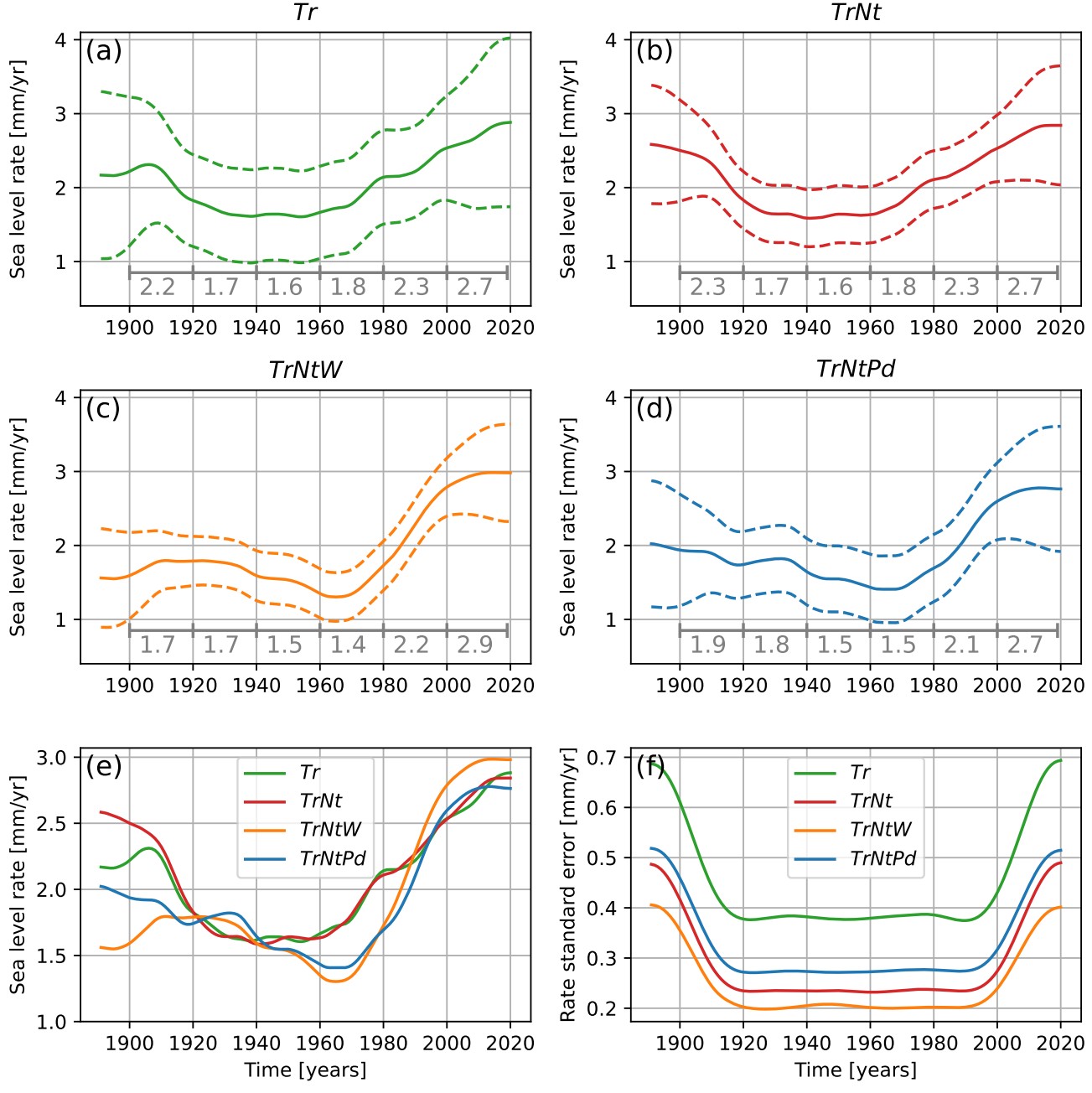

**Figure 4.** (a–d) The rates of SLR were obtained for four different statistical models. The period shown here is 1891 to 2019. The dashed lines show the 5th and 95th percentiles of the uncertainty range computed from a parametric bootstrap method. Numbers in grey under the curves indicate the mean rates for six consecutive 20-year periods ([1900–1919], [1920–1939], [1940–1959], [1960–1979], [1980–1999], [2000–2019]) (e) Median sea level rates. (f) Standard error of the sea level rates.

a constant over an arbitrary period as was done in Calafat and Chambers (2013) and Steffelbauer et al. (2022). In addition, we propose a careful uncertainty analysis accounting for serially dependent unexplained fluctuations, which is used to evaluate the strength of the evidence for an acceleration. These two elements help to avoid framing the problem of acceleration detection as binary. This is important when advising decision-makers: significance testing based on ad hoc models like a broken-line trend may lead to a paradigm shift from a steady rate of SLR in one year to an accelerating rise just years later, as demonstrated by the results in Calafat and Chambers (2013) and Steffelbauer et al. (2022). To our best knowledge, the GAM has not been applied to estimate trends and acceleration in sea-level data before and we believe it could help solve similar acceleration detection problems in regions other than the coast of the Netherlands.

Over recent decades, our best-fitting model yields a rate of SLR of $2.9\,{}^{3.5}_{2.4}$ mm/yr over 2000–2019 (Tbl. 2). This is in agreement with results of Steffelbauer et al. (2022) for the North Sea and by Frederikse et al. (2020) for the North Atlantic who find rates of, respectively, $2.7\,{}^{3.1}_{2.3}$ and $2.7\,{}^{3.3}_{2.1}$ mm/yr over 1994–2018.

When removing the wind influence from the sea-level observations, the underlying assumption is that this influence is only due to natural variability and that there is no structural change due to anthropogenic forcing. However, as we find a wind-driven trend over the entire period of study 1836–2022 from both the wind and pressure difference model (Fig. 3b); this trend could also continue in the future. We do not know of any study investigating the possible cause of such a trend. If it is caused by climate change due to anthropogenic forcing, it would be reasonable to expect it to continue in the future. Conversely, if it is caused by natural variability, it might reverse. Most of the CMIP5 and CMIP6 ensembles do not show a systematic trend associated with wind influence on sea level in the North Sea over the historical period or in future scenarios. So, these models may miss the process driving the trend in the observations or the trend in the observations may be natural variability. In each case, the magnitude of around 0.15 mm/yr over the historical period is small enough compared to other sources of SLR uncertainty to neglect it when making sea-level projections on time scales of more than several decades.

The four GAMs indicate a decrease in the rate of SLR from the beginning of the 20th century until about the 1960s, with a minimum in the 1940s for *Tr* and *TrNt* and in the 1960s for *TrNtZw* and *TrNtPd* as can be seen in Fig. 4. The decreasing rate of SLR as seen in Fig. 4 could be due to the strong Arctic warming from 1900 to 1930, followed by an Arctic cooling from 1930 to 1970 (Fig. 4, Bokuchava and Semenov (2021)). This could have influenced the rate of SLR by reducing the glacier loss rate or decreasing the local steric change. Since the local sea-level budget is not closed before 1950 (Frederikse et al., 2020), we can only speculate about the causes of the drop in the rate of SLR.

From daily to the interannual time scales, the wind influence on sea level in shallow seas is well understood through barotropic theory of the interplay between the Coriolis force, pressure gradient and surface wind stress (equation 3 from Mangini et al. (2021)). On multidecadal time scales, it is possible that the physical mechanism underpinning the relation between wind and sea level also involves steric sea level change (Chen et al., 2014; Dangendorf et al., 2021). In particular, baroclinic signals in the deep ocean propagate as a volume flux into shallow seas (Bingham and Hughes, 2012; Calafat et al., 2012). However, since the regression coefficients to obtain the wind influence on SLR are determined using the annual data including the large interannual variability (see the large spectral power of the wind influence estimates in Fig. 3c), we think these coefficients mostly reflect the barotropic wind influence.

We find a strong increase in the rate of SLR between the 1960s and 2000s (Fig. 4). Based on accelerating globally-averaged SLR, we expect that the SLR acceleration at the Dutch coast continues beyond the year 2000. However, the standard error of the rate of SLR increases due to the approaching end of the time series and therefore, our method cannot say for certain whether the acceleration persists. One potential application of the reconstructed SLR evolutions would be the extrapolation of the observed rate into the near future. This method was recently used as an additional line of evidence for future sea-level rise by Sweet et al. (2022). Based on Fig. 4c model *TrNtW*, we could assume a constant rate of 2.8 mm/yr from 2000 onwards, arriving at a rise of 0.28 m between 2000 and 2100. However, also a constant acceleration can be assumed. The difference in rate between 1975 and 2000 is 1.3 mm/yr which gives a constant acceleration of $1.3/25 = 0.05$ mm/yr$^2$ (as inferred from the trend in sea level of Fig. 4c), resulting in a rise of 0.5 m, from 2000 to 2100, which is much higher than the rise without acceleration. However, given the complexity of changes in the various drivers of global SLR, it would be naive to assume that the acceleration will remain constant during the remainder of this century. Nonetheless, these crude extrapolations illustrate the practical significance of our estimates of the local rates of SLR and the importance of obtaining the evolution of these rates over time.

In the appendices, we show and discuss our nodal tide estimates (App. A), the rates of SLR of the individual tide gauge stations (App. B), and the relationship of the multidecadal wind influence on sea level with two well-established modes of variability in the North Atlantic, the North Atlantic Oscillation and the Atlantic Multidecadal Variability (App. C). In App. A, we show that the estimates of the nodal tide in *TrNt*, *TrNtW* and *TrNtPd* have amplitudes of more than 2.5 times the amplitude of the equilibrium tide and they lead the equilibrium tide by 3 years. However, only correcting the sea level using the equilibrium tide leaves a large amount of spectral energy close to the period of the nodal tide. Our hypothesis for the deviation from equilibrium tide along the Dutch coast, which needs more extensive research, concerns the steric sea level: non-linear dynamics of the nodal tide inside the North Sea basin could drive vertical-mixing processes that drive steric sea level. In a budget study (as was done by Frederikse et al. (2016)), these nodal-driven effects are classified as steric effects in the budget, making the equilibrium tide successful in closing the budget. In App. B, we show the rates of SLR for the individual tide gauge stations using the GAM *TrNtW*. The rates of SLR for the individual stations show large differences which could be due to unaccounted-for vertical land motion, tidal effects, large-scale engineering projects affecting coastal dynamics, or measurement errors, especially further in the past (Baart et al., 2019). Additional research is needed to better understand which physical processes drive the differences in the local sea-level rates.

## 6  Conclusions

In this study, we estimate the sea-level trend and the influence of the nodal tide and wind on sea level along the coast of the Netherlands. We analyse the average of the observations from six tide gauges and zonal and meridional wind and atmospheric pressure at sea level from two reanalysis data sets. Using four different GAMs, we estimate a smooth trend and (depending on the model) the effects of the nodal tide and wind. One model has no predictive variables; others have only nodal tide or additionally include zonal and meridional wind or pressure gradient as predictive variables. We find that using the local zonal

and meridional wind as predictive variables best estimates the sea-level trend based on the reduction of the deviance as well as the standard error. The deviance is reduced when more predictive variables are added to the GAM: by 11% when adding the nodal tide and by another 33 to 52% when adding the wind forcing.

Estimating the wind influence based on different choices of predictive variables in *TrNtW* and *TrNtPd* shows the method's robustness, as both models lead to similar conclusions. We find a long-term sea-level rise due to wind forcing of 0.13 mm/yr or 0.14 mm/yr for 1929–2022, depending on the choice of model (Fig. 3b). The long-term strengthening of the wind is consistent with an observed northward shift and strengthening of the jet stream (Hallam et al., 2022). Also, we find a low-frequency wind variability which can rise or drop sea level by about 1 cm over 2 to 5 decades (Fig. 3d). Using a coherence analysis we relate this variability to both the North Atlantic Oscillation and the Atlantic Multidecadal Variability (App. C). Using the GAMs *TrNt*, *TrNtW* and *TrNtPd* we obtain estimates of the nodal tide with amplitudes of more than 2.5 times the amplitude of the equilibrium tide as well as a phase leading the equilibrium tide by 3 years (App. A).

After obtaining the sea-level trend using the four GAMs, we obtain the rate of SLR by differentiating the trend. This results in new insight into the evolution of the rate of SLR along the coast of the Netherlands over the observational period (Fig. 4. The rates of SLR, excluding the influence of the wind, are lower at the beginning of the 20th century and larger at the beginning of the 21st century. Our best-fitting model yields a rate of SLR, excluding nodal and wind effects, of $2.9\,^{3.5}_{2.4}$ mm/yr over 2000–2019 compared to $1.7\,^{2.3}_{1.3}$ mm/yr in 1900–1919 and $1.5\,^{1.9}_{1.2}$ mm/yr in 1940–1959 (Tbl. 2). The probability (the p-value) of finding a rate difference between 1940–1959 and 2000–2019 equal to the one we found when there would not have been an acceleration is smaller than 1% (Tbl. 3). These results provide a clear indication of an acceleration of SLR. Also, we find, for the first time, that the acceleration of SLR along the coast of the Netherlands started in the 1960s. This aligns with global SLR observations and expectations based on a physical understanding of SLR related to global warming (Fox-Kemper et al., 2021; Dangendorf et al., 2019). Furthermore, we explain that the acceleration of SLR along the Dutch coast has been difficult to detect due to the masking of the acceleration by wind-field and nodal-tide variations.

*Code and data availability.* All data and code used in this study are available in the GitHub repository at: https://github.com/KNMI-sealevel/NetherlandsSeaLevelAcceleration. Additionally, the code and data are deposited on Zenodo with the identifier https://doi.org/10.5281/zenodo.7994728 (Keizer et al., 2023). We use the following datasets: PSMSL tide gauge data (https://psmsl.org/, last accessed: 2023-02-01); reanalyses: 20CR (Slivinski et al., 2019), ERA5 (Hersbach et al., 2020), COBE-SST (Hirahara et al., 2014), HadISST (Rayner et al., 2003); climate indices: NAO (Jones et al. (1997), (https://crudata.uea.ac.uk/cru/data/nao/nao.dat, last accessed 2023-03-09), AMV (Enfield et al. (2001), http://www.psl.noaa.gov/data/timeseries/AMO/, last accessed 2023-03-09).

# Appendix A: Nodal Effects on Sea Level

The nodal effects on sea level are represented by the second term of the equations shown in Tbl. 1 of our GAMs *TrNt*, *TrNtW* and *TrNtPd*. Figure A1a shows the estimates of the nodal tide from different GAMs, as well as the equilibrium tide. The equilibrium tide is obtained for each of the six tide gauge stations whereafter their average is obtained (Woodworth, 2012).

We find that the nodal tide amplitude is $1.44$ cm, $1.45$ cm and $1.35$ cm for respectively *TrNt*, *TrNtW* and *TrNtPd* compared to an amplitude of $0.54$ cm for the equilibrium tide. We also find that their phases lead the phase of the equilibrium tide by 3 years. Figure A1b shows the spectra of the residual obtained by subtracting the reconstructed sea level from the observed sea level as well as the spectra of the equilibrium tide. The reconstructed sea level is obtained for a model *TrW* which includes the sea-level trend and zonal and meridional wind but no nodal tide, for the model *TrNtW* and for a model *TrEtW* where the predictive variables are the same as for *TrW* but the sea level data is corrected using the equilibrium tide. Since the models *TrEtW* and *TrNtW* only differ in the way the nodal tide is obtained, we can study the effect of the method on the resulting nodal tide estimation. The spectra are obtained using a multitaper method (Lees and Park, 1995). Due to the use of the multitaper method, the spectrum of *Et* is not a single line at 18.613 years but rather a peak centered around that period as a result of the windowing. As expected, around the period of the equilibrium tide, most energy remains in the residuals of *TrW* as this model does not include the nodal tide leaving the nodal tide signal in the residuals. When the equilibrium tide is included in the model (*TrEtW*), the nodal tide signal should no longer be included in the residuals and therefore the spectrum should contain less power around the period of the equilibrium tide. We see that indeed less power remains around this period and that the removed power is equal to the power of the equilibrium tide (*Et*). However, a lot of energy remains compared to using *TrNtW*. This result underlines our choice to use a statistical estimation for the nodal tide.

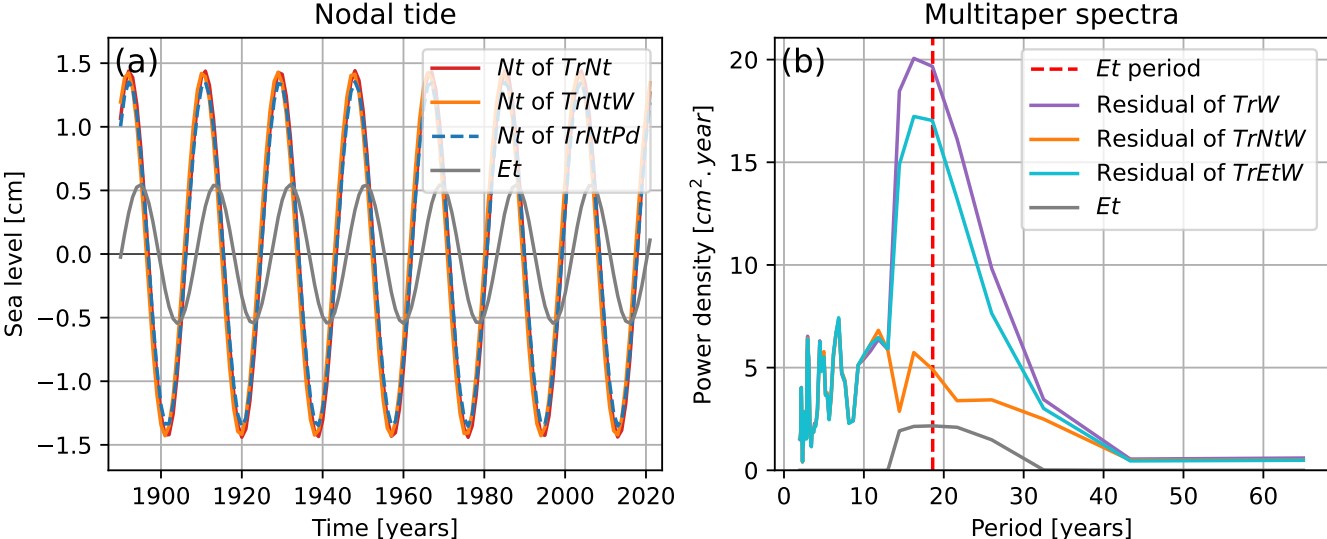

**Figure A1.** (a) Comparison of the influence of the nodal tide on sea level resulting from the GAMS *TrNt*, *TrNtW* and *TrNtPd* to the equilibrium tide. (b) Spectra of the residuals for a model including only trend and wind (*TrW*), including trend, nodal cycle and wind (*TrNtW* and including only trend and wind but with sea level corrected for the equilibrium tide (*TrEtW*). The residuals are obtained by removing the estimated sea level from the observed sea level. Also, the spectrum is shown for the equilibrium tide (*Et*) as well as the equilibrium tide period at 18.613 years (Et period).

## Appendix B: Rates of SLR for individual tide gauge stations

In this study, we have used the average of the six tide gauges along the Dutch coast (Fig. 1a) to estimate the rate of SLR while reducing the influence of local processes. We obtain the rates of SLR and their standard errors for each tide gauge station (Fig. B1). We use *TrNtW* as the sea-level rates for the average of the six stations resulting from this GAM have the lowest standard error (Fig. 4f). The standard errors of the rates (Fig. B1b) are mostly higher for the individual tide gauges than for their average. The rates of SLR for the individual stations show large differences, in particular in the first half of the

observational period. Before the 1960s, the spread in sea-level rates between the stations is larger than after; while the rates of some stations are increasing, for other stations they are decreasing. After the 1960s, the rates for most stations show an overall increase as well as a smaller spread between the stations.

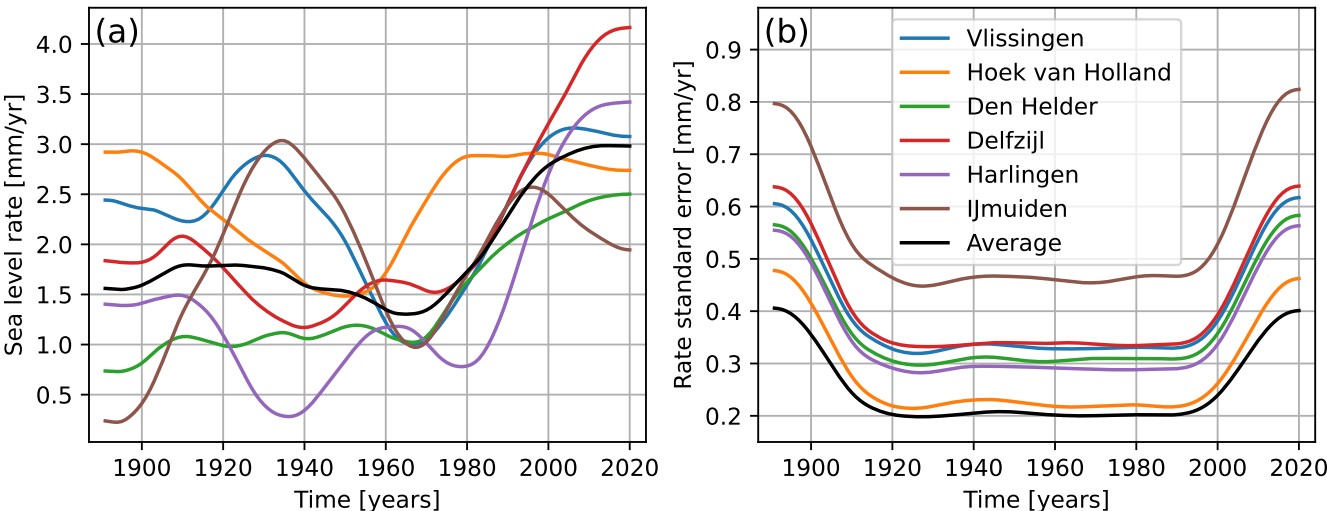

**Figure B1.** The rates of SLR obtained per tide gauge station using the GAM *TrNtW* (a) The rates of SLR per tide gauge station as well as their average obtained from *TrNtW*. (b) Standard error of the sea level rates.

## Appendix C: Multidecadal sea-level variability

In Fig. 3d, we found that our two estimates of wind influence on Dutch sea level exhibit multidecadal variability with an amplitude of about 1 cm and a period of 2 to 5 decades. This multidecadal wind influence estimate was derived by removing third-order polynomial fits of the *W* and *Pd* components of the *TrNtW* and *TrNtPd* GAMs, respectively (Figs. 3b), and subsequently applying a 21-year LOWESS filter (Fig. 3c). Previous studies have not revealed this wind-driven low-frequency sea-level variability in Dutch sea-level observations. As our two wind influence estimates are based on the wind at the Dutch

coast and sea level pressure difference over Europe, respectively, it stands to reason that they are related to the large-scale North Atlantic climate state and its internal variability. There are two well-established North Atlantic modes of variability, the North Atlantic Oscillation (NAO), measured by the pressure difference between the Iceland Low and the Azores High, and the Atlantic Multidecadal Variability (AMV), measured by the North Atlantic sea surface temperature (SST) anomaly. In this paragraph, we analyse the relation of our low-frequency wind-influence estimates to North Atlantic SSTs as well as to indices

of the NAO and AMV. We also discuss possible mechanisms for these relationships.

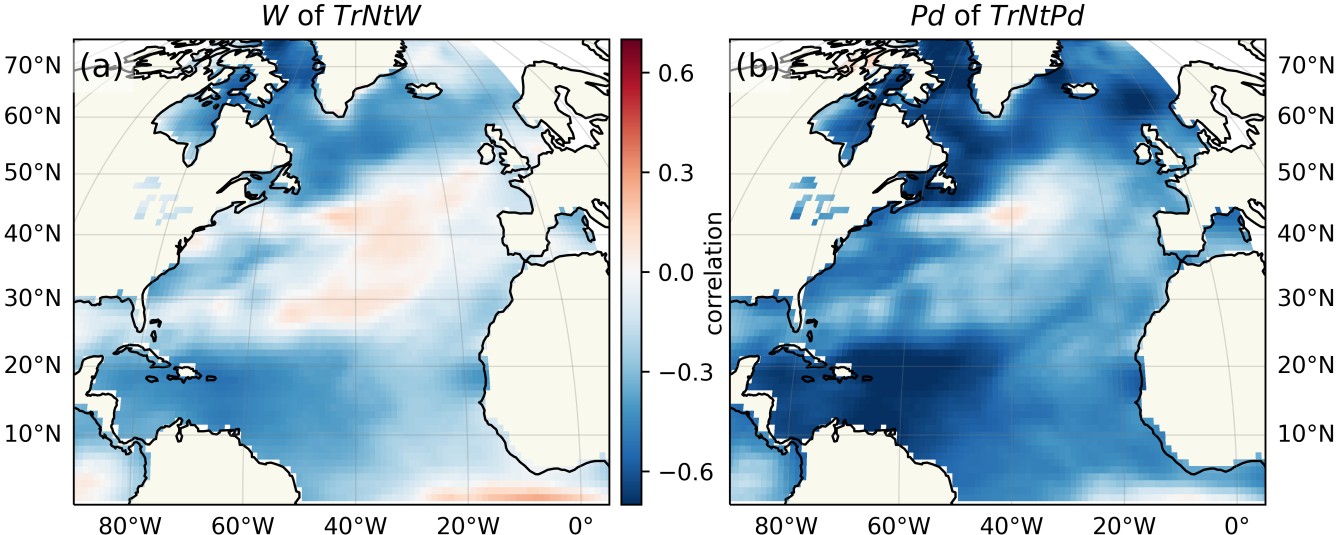

**Figure C1.** Correlation pattern of our multidecadal wind-influence estimates, *W* of *TrNtW* (a) and *Pd* of *TrNtPd* (b), with the North Atlantic sea surface temperature field. Both the wind-influence time series as well as the SSTs at each geographic point are detrended with a third-order polynomial and smoothed with a 21-year LOWESS filter (see Fig. 3d for the detrended and smoothed wind-influence time series).

With Fig. C1, we focus on the correlation of our wind-influence estimates with North Atlantic SSTs. Depending on time scale, patterns of anomalous SSTs can be both a cause and a consequence of anomalous winds through various mechanisms of air–sea interactions. On short time scales, atmospheric variability determines North Atlantic SSTs while on multidecadal time scales, the oceanic heat convergence drives the North Atlantic SST signal (Woollings et al., 2015). The NAO imprints on the

395 SST on inter-annual time scales in a tripole pattern, while on multidecadal time scales the SST anomalies influence the NAO behaviour, in particular its persistence behaviour. The AMV index measures the average North Atlantic SST anomaly. Using the COBE-SST2 reanalysis from 1850–2019 (Hirahara et al., 2014), we correlate the low-frequency wind influence estimates of Fig. 3 with the similarly detrended and smoothed North Atlantic SST field. We see generally negative correlations with a tripole pattern with more negative correlations in the north and south and neutral or even positive correlations in the central

North Atlantic. Of particular interest is the meridional SST gradient around 50°N visible through the correlation gradient in Fig. C1 which affects the zonal wind around this latitude.

The NAO is a mode of atmospheric variability that influences, among others, the storm tracks and hence average wind over the North Atlantic and the North Sea. The NAO is known to influence the sea level in the North Sea, especially in winter (Jevrejeva et al., 2005; Dangendorf et al., 2012, 2014a). In atmosphere–ocean general circulation model simulations, Dangendorf et al. (2014b) found a statistically significant relationship between the NAO and atmospherically induced mean sea level changes in the German Bight. For our analysis, we use the annual NAO reconstruction by Jones et al. (1997) which covers the period 1825–2021 and measures the pressure difference between Gibraltar and southwest Iceland. The AMV measures the multidecadal variability of the North Atlantic SSTs and is connected to changes in the Atlantic Meridional Overturning Circulation. We use the annual AMV index time series starting in 1856 provided by the Physical Sciences Laboratory of the United States National Oceanic and Atmospheric Administration (Enfield et al., 2001).

Figure C2 shows time series, spectra, coherence and phase difference of the wind influence estimates, *W* of *TrNCW* and *Pd* of *TrNcPd*, together with indices of the NAO and the AMV. Panels (a) and (b) show the annual, standardised and 21-year LOWESS smoothed time series. Panels (c) and (d) show the multitaper spectral estimates of these time series (like Fig. 3c). The wind influence and NAO spectra are approximately white, i.e. having similar spectral power at all periods, while the AMV time series is clearly red with spectral power concentrated at multidecadal periods. The third row shows the coherence spectra between pair-wise combinations of the time series, while the estimated phase difference is shown in the last row. The highest coherence is observed between the two wind-influence time series, *Pd* and *W*, except at periods close to 20 years with approximately zero phase lag, suggesting that they measure the same multidecadal wind influence on sea level. Both wind influence time series show medium coherence with the NAO, peaking between 20 and 30 years, with little phase difference at periods longer than 20 years. The coherence with the AMV is high for both wind-influence time series being anti-phase at multidecadal periods, especially after longer than 30 years, meaning higher North Atlantic SSTs correlate with lower wind-induced sea levels at the Dutch coast on these multidecadal time scales. The NAO and AMV are anti-correlated, especially at periods longer than 10 years, though their coherence is low, a finding consistent with the study by Klavans et al. (2019). We also investigated the effect of limiting the time series length to more recent time, e.g. from 1890, where qualitatively the same relationships hold (not shown).

The picture that emerges from the coherence analysis in Fig. C2 is that the NAO is positively correlated with the wind influence on Dutch sea level, especially around periods of 20–30 years and the AMV is negatively correlated, in particular at periods longer than 30 years. The out-of-phase NAO–AMV relationship (Fig. C2e) has been found and studied previously (e.g., Peings and Magnusdottir, 2016). Even though *Pd* of *TrNcPd* reflects a meridional atmospheric pressure gradient similar to the NAO (albeit shifted eastward), the relatively low *Pd*-NAO coherence (Fig. C2f) suggests that the NAO is an inferior proxy for annual sea-level variability along the Dutch coast compared to the pressure gradient of *Pd* (Fig. 1b) The overall negative correlation with the smoothed SSTs of Fig. C1 is also expressed as an out-of-phase relationship between the wind influence estimates and the AMV (Fig. C2g/h). Strengthening of the meridional SST gradient around 50°N strengthens the meridional pressure gradient and hence the zonal westerly winds which increases the wind-driven sea-level signal (Hallam et al., 2022). Furthermore, the negative SST correlation north of South America is related to a shift in the Intertropical Convergence Zone

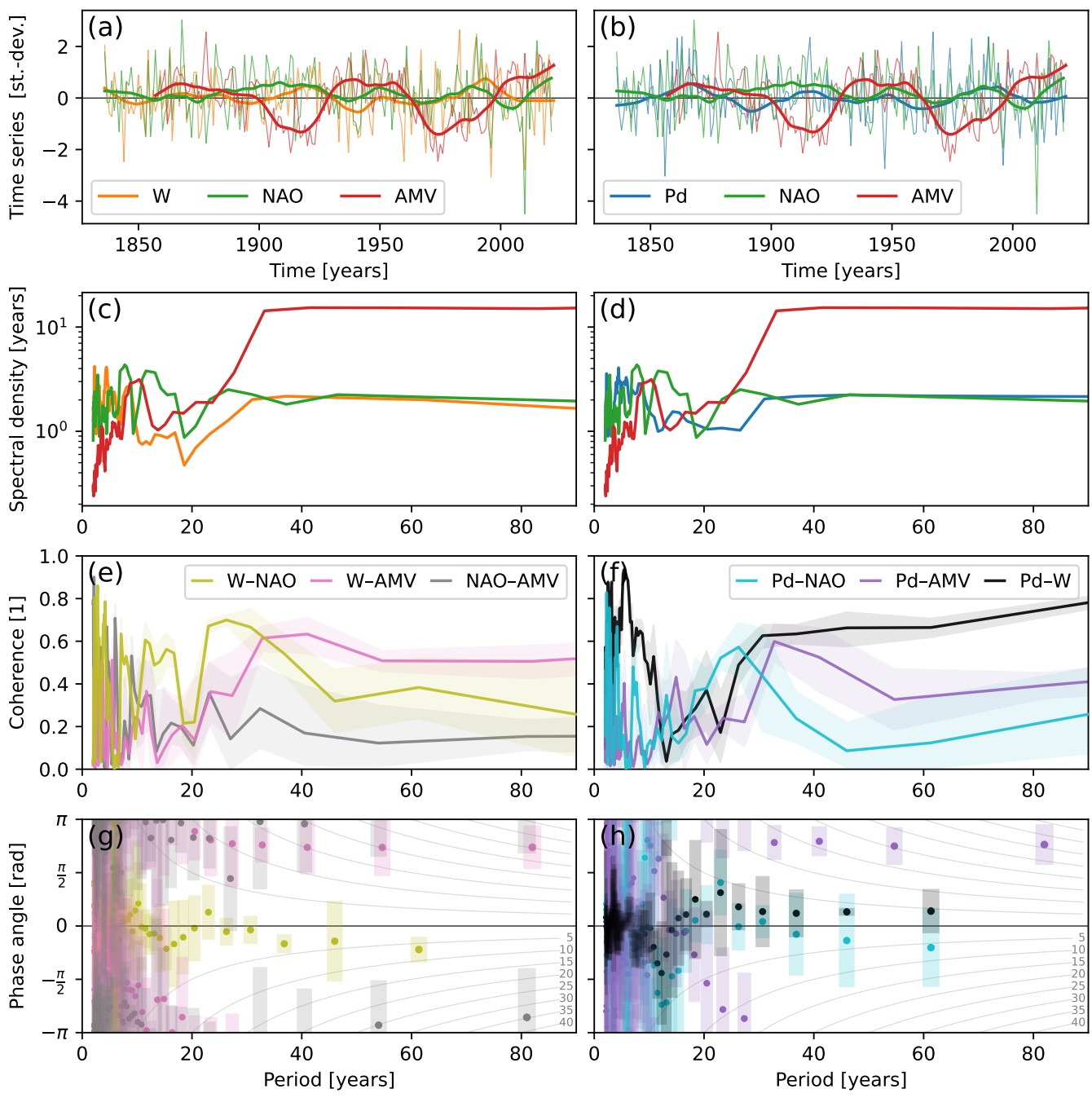

**Figure C2.** Time series analysis of the wind influence estimates, *W* of *TrNCW* (left) and *Pd* of *TrNcPd* (right), and indices of the North Atlantic Oscillation (NAO) and Atlantic Multidecadal Variability (AMV). (a/b) The annual, unit standard deviation (st.-dev.) time series (thin lines) together with their 21-year LOWESS-filtered versions (thick lines). (c/d) Multitaper spectral estimates of the annual time series. (e/f) Spectral coherence estimates of pairs of time series with 5-95 percentile uncertainty range shaded. (g/h) Phase shift estimates with uncertainties as error bars. The grey curved lines in the background with labels ranging from 5–40 translate the phase shift in radians to years at each frequency with a positive (negative) phase denoting the first time series leading (lagging) the second time series.

which triggers an eastward-tilting atmospheric Rossby wave train affecting wind speeds over Central Europe (Okumura et al., 2001).

Naturally, there are limitations to this exploratory analysis. We only investigated annual time series and neglected the seasonality of the effects, though we focus here on multidecadal time scales. The time series are also relatively short compared to the multidecadal time scales of interest which affects spectral estimation in particular. Furthermore, all observed climate variables used here are subject to anthropogenically-forced trends. Removing these trends is necessarily imperfect; we have used cubic polynomial detrending for the wind-influence estimates and the North Atlantic SSTs, and the AMV time series is only linearly detrended. To investigate whether the findings are influenced by our choice of SST reanalysis dataset, we also performed the SST correlation analyses of Fig. C1 with the HadISSTv1.1 SST reanalysis of the Met Office Hadley Centre (1870–2021; Rayner et al. (2003)) and confirmed that the results are very similar (not shown). Despite these limitations, we can conclude that the sea-level variability at the Dutch coast at multidecadal time scales is influenced by both the NAO and the AMV, though more research is needed.

*Author contributions.* Iris Keizer, Dewi Le Bars, André Jüling and Cees de Valk developed the model code and performed the simulations. All authors contributed to the interpretation of the results. Iris Keizer and Dewi Le Bars prepared the manuscript with contributions from all authors.

*Competing interests.* The authors declare that they have no conflict of interest.

*Acknowledgements.* This publication was supported by the project RECEIPT (REmote Climate Effects and their Impact on European Sustainability, Policy and Trade) which received funding from the European Union's Horizon 2020 Research and Innovation Programme under Grant Agreement No. 820712 and by PROTECT a European Union's Horizon 2020 research and innovation program under Grant Agreement No. 869304.

Iris Keizer, Dewi Le Bars and Sybren Drijfhout gratefully acknowledge support from the Netherlands Knowledge Programme on Sea-level rise that supported them with a special grant. Roderik van de Wal and André Jüling acknowledge support from the Netherlands Polar Program to the Dutch Polar Climate and Cryosphere Change Consortium under File No. ALWPP.2019.003.

In this study, we used the GAM implementation, Lowess filtering and third-order detrending tool from the statsmodel library (https://www.statsmodels.org/; Seabold and Perktold (2010)), the multitaper method from the spectrum library (https://pyspectrum.readthedocs.io), the ordinary least squares regression model from the scikit-learn library (https://scikit-learn.org) and the multitaper method from the mtspec library (https://krischer.github.io/mtspec/).

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
