# Peer review of "The acceleration of sea-level rise along the coast of the Netherlands started in the 1960s"

_EGUsphere, 2022_

## Referee Comment (RC1)

Review of "The acceleration of sea-level rise along the coast of the Netherlands started in the 1960s" by Keizer et al., 2022 for Ocean Science

**Summary**

In this manuscript, Keizer and colleagues use a new statistical method (GAM) to infer sea-level trends and identify acceleration in the time series. Based on six tide gauges along the Dutch coast, they identify that SLR has been accelerating since 1960s, and that this acceleration is masked by wind influence on sea level. Their study highlights not only the importance of including the influence of wind and of the nodal cycle on sea level, but also show how GAM can be used to infer on the rate of sea level in other locations. This is a relevant study, that deserves to be published. I do think some things should be clarified or better presented before final publication.

**General Comments**

- Periods used: It was unclear to me how the authors chose the periods shown in Figure 3 and Figure 4. For example, in Figure 3b, the trend until 1928 and for 1928-2020. This split doesn't coincide with the splits in Figure 4, and neither with the different periods of the two atmospheric reanalyses used. And in Figure 4, the authors give trends for 4 periods of 20 years, but it was not explained why 20 years, and why those specific divisions also. For example, why no trend over 1959 to 2000? Would be interesting to see the trend of the acceleration.

- I think would be good to add a table to the manuscript, with the trends for the different GAM models and the different periods. Right now, the trends are given in Figure 4, and through some places in the text, but would be easier to follow in a table. And would also be good to know the trend for the entire period (1900-2019) for comparison, and a trend for before the acceleration (before 1960s) and after, not only on 20 years interval.

**Line-by-line comments (Specific comments)**

- Abstract L4: "covering the period 1890-2000". I think it was supposed to be until 2019. As it's mentioned in L11 and in the rest of the manuscript.

- Figure 3: It was unclear to me if these are trends of wind, wind influence on sea level (which would be just sea-level trends), or just sea-level trends.

- L16: This line should be part of the previous paragraph (usually abstracts are a single paragraph).

- L22: sea-level rise, with hyphen (to be consistent with the fact that you always used a hyphen when sea level was a modified (e.g., in the same line "sea-level projections")

- L26: Should add that the contribution of Greenland is much smaller than the globally averaged contribution to the North Sea, it's not the case for most of the Southern Hemisphere (e.g., see Figure 4 of Camargo et al., 2022).

- L50: Should change the order of the verd: "have been used as predictive variables by various authors"

- L80: Would be good to refer here to Figure 1a.
- L80: I was also wondering why you didn't include Maassluis tide gauge, like in Steffelbauer et al (2022).
- L107: (Wood, 2020) not Wood (2020)
- L155: "see Section 2 of Cox and Reid (2004)" should be between parathesis.
- Figure 2: I missed a bit more of discussion about Figure 2, there are only 4 lines about it... Also, why was TrNcPd time series excluded from this figure? I think would be good to show the time series for the 4 GAMs, or at least explain why you decided not to show it.
- L185-186: It was unclear if the authors actually tested if the increased degrees of freedom (DoF) increase the standard error of the trend enough to justify that using only one predictive variable for the wind is better than using two. Table 1 seems to suggest that the increasing the DoF is worth it, since it gives a significant decrease in the deviance. So it seems a bit contradictive then to use the increased DoF as a reason to not use more than once wind variable. I was curious if the TrNcPd model would perform better than with TrNcZw, if you had used both boxes as proxies, as suggested in Dangendorf et al (2014b), instead of taking the difference between the boxes as it was performed here.
- L195-196: Why are your results in contraction with the ones of Dangendorf et al (2014a) Do you mean that they get a negative trend while you have a positive one in Figure 3b? Are you then comparing your trends from 1836-1928 and 1929-2020 with the ones from 1953-2003 and 1900-2011 from Dangendorf et al (2014a)? If that is the case, it's not very consistent to compare trends of such different periods. Then the difference in the results would not be only the results of an update in the atmospheric reanalysis, but also from the fact that you are comparing different periods.
- Section 4.3: Was a bit confusing how you start with Figure 4, discusses Table 2, and then moves back to Figure 4. I would suggest have it clearly separated, to be less confusing for the reader (and maybe moving the paragraph starting in L231 to before L215, to introduce the acceleration in the 1960s).
- L210-212: Would it be possible to give a number when you talk about lower/higher uncertainties here? Maybe the average width of the confidence interval for the time series?
- L212: Refer here to Figure 4f.
- L222-227: You talk here in percentages, and table 2 is in probability. Just a suggestion, but maybe you should have Table 2 in % as well, to be consistent.
- L228-230: This is a very important conclusion of your results. You should emphasize this in the conclusion (sorry if you have done it and I missed it).
- L231: typo: 1960".
- Figure 3 caption: would be good to add to the caption the line colors as well: "(TrNcZw, orange line) … (TrNcPd, blue line)". Also, when referring to Figure 1, should be Figure 1b.
- Table 2 caption: I guess this is more of a style choice, but I would say that the explanation that starts with the "For example, …" should be in the main text, and not in the caption of the table.

- L241: "Calafat and Chambers (2013) and Steffelbauer et al (2022).
- L246: "results in Calafat and Chambers (2013) and Steffelbauer et al (2022)." (not between parenthesis).
- L259-262: This is a very long sentence (4 sentences) … Also, 1 paragraph sentences should be avoided.
- L267: Maybe also good to refer to the work from Dangendorf et al (2021) here?
- L275: Is the 1.5 the rate of the acceleration period (1975-2000)? That is not shown in Figure 4c. Here also would benefit from having a table with the trends for the different periods (as suggesting in General Comment 2). Also the reason why you divided by 25 (I assume is the number of years from 1975-2000) was not super clear, it would be good to have this better explained, as this is a very important application of the results.
- L285-286: This sentence implies that you did test the model using more than one predictive variable for the wind (which was unclear before). If you did test it, I would suggest having these results in the supplementary information, so that the reader can actually see it.

**References:**

Camargo, C. M. L., Riva, R. E. M., Hermans, T. H. J., & Slangen, A. B. A. (2022). Trends and uncertainties of mass-driven sea-level change in the satellite altimetry era. *Earth Syst. Dynam. Discuss.*, *13*, 1351–1375. Retrieved from https://esd.copernicus.org/preprints/esd-2021-80/

Dangendorf, S., Frederikse, T., Chafik, L., Klinck, J. M., Ezer, T., & Hamlington, B. D. (2021). Data-driven reconstruction reveals large-scale ocean circulation control on coastal sea level. *Nature Climate Change*, *11*(6), 514–520. https://doi.org/10.1038/s41558-021-01046-1

---

## Referee Comment (RC2)

Review of the manuscript: "**The acceleration of sea-level rise along the coast of the Netherlands started in the 1960s**" by Iris Keizer et al.

The paper estimates the sea-level rise (SLR) using the average of six tide gauges along the coast of the Netherlands, from 1890 to 2020. For this purpose, the authors used four Generalized Additive Models – GAMs, to estimate the sea-level trend, as well as the influence of the lunar nodal cycle and zonal wind on sea level, the latter using two different approaches. Results indicate an acceleration of SLR starting in the 1960's, which protrudes once the tidal and wind effects on sea level are removed. Besides, they show that wind effects force a long-term SLR, as well as a low-frequency variability modulating sea level by about 1 cm, which is related to Sea Surface Temperature variations in the North Atlantic.

I find the aim of the paper very relevant, as the assessment of coastal rates of SLR and its acceleration due to anthropogenic forcing, is essential to obtain adequate projections of future sea levels, necessary to perform effective coastal adaptation strategies. Besides, I find the paper scientifically relevant, especially for the use of GAMs to estimate the sea-level trends, allowing them to isolate the influence of particular variables.

The paper is well written and has an excellent presentation quality. However, there are few comments I would like the authors to consider before submitting this preprint for publication in a scientific journal.

Main comments:

L103. Through the entire paper, the assessment of the sea-level along the coast of the Netherlands uses the average of the six tide gauges. I suggest to include a comment about the reasons of using this average instead of using the individual time series (or both). Furthermore, I think it would be interesting to assess if the SLR trends and acceleration observed in the averaged time series stand, if the same method is applied to the individual tide gauge time series. In particular, the wind effect on sea level could be very different from one station to the other, depending not only in spatial differences in the wind forcing, but also due to coastline configuration, bathymetric differences, among others.

L112. Two sinusoidal waves in opposition of phase with the 18.6 year period were used to assess the nodal effect on sea-level. Although in L120 the authors recognize this approach might remove some additional variability around the nodal period, I suggest the authors to include in the results section, at least one paragraph describing the amplitude and phase lag of the nodal cycle clearly seen in the TrNc GAM. This result should be compared to what is known about this cycle in literature.

Appendix A. In L321 authors indicate they assess the low-frequency relation between wind influence on sea level and low-frequency SST. At this point I think authors should guide the reader, indicating the physical relation between these two

variables they are trying to expose. Later in L345, authors offer a physical explanation for the relation found between SST in the north box and zonal wind. In my view, the hypothesis that changes in the meridional temperature gradient strength the jet stream, is not strongly presented. Due to air-sea interaction, SST has an inverse relationship with atmospheric pressure at sea level. As used in the paper (TrNcPd), changes in the meridional atmospheric pressure gradient modulate zonal wind in the region. Therefore, the relation found between wind influence on sea level and SST is probably possible due to variations in the atmospheric pressure gradient, what in the region is measured by NAO. This reasoning also supports the stronger correlations found in TrNcPd when compared to TrNcZw. I suggest authors to review literature about the relation between NAO and SST (air-sea coupling) in the North Sea (e.g. doi/10.1029/2022JD037270), to present a stronger case in the Appendix. Changes in the Appendix might force some changes in the main paper.

Specific comments:

L4. …covering the period 1890-2020.

L13. Verify a typing error.

L134. Authors assess sea level from 1890 to 2020. Suddenly in this line, they mention that wind effects on sea level are assessed from 1836 to 2020. I suggest authors to include a comment about the reasons of extending in time the assessment of the wind forcing.

L189. In this line authors assess the wind-driven trend in sea level for the 1928-2020 period. However, Figure 3b show trends for two periods. I suggest authors to include a comment about the wind-driven sea-level trends observed during the first period. I think this is important especially due to the large difference in the trends observed in TrNcPd.

L232. The authors speculate about the reasons behind the SLR decreasing rate observed from 1900 to 1960. Please consider to move this explanation to the discussion section.

Table 2 legend. The probability in the third line is 0.23.

L246. Verify the referencing.

L274. 2000 onwards, …

Figure A1 legend. Define the AMV acronym, indicating that the area is shown with the black limits. Try to use colors that can be easily distinguished in panels' b and d.

---

## Author Comment (AC3)

**DOI:** 10.5194/egusphere-2022-935
**Version:** Revision
**Title:** The acceleration of sea-level rise along the coast of the Netherlands started in the 1960s
**Authors:** Iris Keizer, D. Le Bars, C. de Valk, A. Jüling, R. van de Wal, and S. Drijfhout

**Point by point reply to reviewer #1**

March 3, 2023

We thank the reviewer for their careful reading and beneficial comments on the manuscript and address their points below. Since we added a table to the revised manuscript (Table 2), Table 2 of the old version is Table 3 in the revised version. Also, two appendix sections are added (Appendix A and Appendix B), making Appendix A of the old version Appendix C of the revised version. Some major changes have been made to the manuscript. We obtained the most recent available tide gauge records and atmospheric data. Therefore, the time series are extended to 2021 and, respectively, 2022. We checked whether using more predictive variables in our models including wind as a predictive variable improved the results. We found that adding the meridional wind to the model that only included the zonal wind improved the results. Therefore, we now include a local wind model that includes both the zonal and meridional wind. We included results on the nodal tide influence in the appendix as well as the sea-level rates for the individual tide gauge stations.

**Summary**

*In this manuscript, Keizer and colleagues use a new statistical method (GAM) to infer sea-level trends and identify acceleration in the time series. Based on six tide gauges along the Dutch coast, they identify that SLR has been accelerating since 1960s, and that this acceleration is masked by wind influence on sea level. Their study highlights not only the importance of including the influence of wind and of the nodal cycle on sea level, but also show how GAM can be used to infer on the rate of sea level in other locations. This is a relevant study, that deserves to be published. I do think some things should be clarified or better presented before final publication.*

**General Comments**

- *Periods used: It was unclear to me how the authors chose the periods shown in Figure 3 and Figure 4. For example, in Figure 3b, the trend until 1928 and for 1928-2020. This split doesn't coincide with the splits in Figure 4, and neither with the different periods for the two atmospheric reanalyses used. And in Figure 4, the authors give trends for 4 periods of 20 years, but it was not explained why 20 years, and why those specific divisions also. For example, why no trend over 1959 to 2000? Would be interesting to see the trend of the acceleration.*

  The time periods that we use to report different trend values are indeed chosen somewhat arbitrarily. However, there are some thoughts behind our choices. The reason to obtain values for trends over certain time periods is to be able to compare those periods. Therefore, we want to split our time series into periods where the trend is rather stable. For Figure 3b, this can be achieved by splitting our time series into two parts. For Figure 4, however, the trends show more changes over time which made us pick periods of 20 years. We will add the 20 years trends over 1960–1979 and 1980–1999. We considered splitting Figure 3b in 20-year periods but considering

that two trends are plotted this would make the figure complex whereas the added informal value is little. On the other hand, using larger periods for Figure 4 would remove valuable information.

- *I think would be good to add a table to the manuscript, with the trends for the different GAM models and the different periods. Right now, the trends are given in Figure 4, and through some places in the text, but would be easier to follow in a table. And would also be good to know the trend for the entire period (1900-2019) for comparison, and a trend for before the acceleration (before 1960s) and after, not only on 20 years interval.*

We agree that adding a table, as suggested, and providing trends over the entire period for Figure 4 provides clarity. Therefore, we will implement the suggestions in the paper. We will also add the suggested trends over longer periods to this table.

**Specific Comments**

- *Abstract L4: "covering the period 1890-2000". I think it was supposed to be until 2019. As it's mentioned in L11 and in the rest of the manuscript*

We should indeed adjust this, we use tide gauge records until 2021, not 2000. (l.3)

- *Figure 3: It was unclear to me if these are trends of wind, wind influence on sea level (which would be just sea-level trends), or just sea-level trends.*

These trends are wind influence on sea level, as is written in the figure's caption and in the label of the y-axis. Since we received other feedback that the terminology 'wind influence on sea level' (which is indeed sea level) is not clear enough, we have decided to add the equations describing our four GAM models to the paper (Table 1). We can refer to these equations and explain which part we are plotting. Also, we updated the titles of the panels of Figure 3 to be clearer about what is plotteed.

- *L16: This line should be part of the previous paragraph (usually abstracts are a single paragraph).*

Indeed, we will adjust this. (l.15)

- *L22: sea-level rise, with hyphen (to be consistent with the fact that you always used a hyphen when sea level was mentioned (e.g., in the same line "sea-level projections")*

Indeed, this will be adjusted.

- *L26: Should add that the contribution of Greenland is much smaller than the globally averaged contribution to the North Sea, it's not the case for most of the Southern Hemisphere (e.g., see Figure 4 of Camargo et al., 2022)*

Thank you, this is mentioned in line (l.36).

- *L50: Should change the order of the verd: "have been used as predictive variables by various authors"*

Indeed, we will adjust this.

- *L80: Would be good to refer here to Figure 1a.*

The comment will be implemented.

- *L80: I was also wondering why you didn't include Maassluis tide gauge, like in Steffelbauer et al (2022).*

  We didn't include the Maassluis tide gauge as we wanted to use the same six stations as the Zeespiegelmonitor, provided by Deltares. The Zeespiegelmonitor is a report on the sea-level rise along the Dutch coast requested by the Dutch government and used for policy choices. Another reason to exclude the Maassluis station is that we want to use six stations that are spatially distributed along the Dutch coast, but the Maassluis station is very close to the Hoek v. Holland station. Furthermore, the Maassluis station is behind a sea barrier (Maeslantkering). Therefore, its sea levels are not similar to the open sea.

- *L107: (Wood, 2020) not Wood (2020)*

  This is improved in the text.

- *L155: "see section 2 of Cox and Reid (2004)" should be between parathesis*

  This is improved in the text.

- *Figure 2: I missed a bit more of discussion about Figure 2, there are only 4 lines about it ... Also, why was TrNcPd time series excluded from this figure? I think would be good to show the time series for the 4 GAMs, or at least explain why you decided not to show it.*

  We included Figure 2 in our paper to guide the reader to our main results in Figures 3 and 4. Showing the time series that result from our GAMs helps to understand what we show in Figures 3 and 4. However, the result of TrNcZw and TrNcPd are very similar. Therefore, we don't think adding the TrNcPd time series improves the paper. We don't intend to expand our discussion of Figure 2 and add TrNcPd as we don't consider this important enough, as we'd also like to keep the paper brief.

- *L185-186: It was unclear if the authors actually tested if the increased degrees of freedom (DoF) increase the standard error of the trend enough to justify that using only one predictive variable for the wind is better than using two. Table 1 seems to suggest that the increasing the DoF is worth it, since it gives a significant decrease in the deviance. So it seems a bit contradictive then to use the increased DoF as a reason to not use more than once wind variable. I was curious if the TrNcPd model would perform better than with TrNcZw, if you had used both boxes as proxies, as suggested in Dangendorf et al (2014b), instead of taking the difference between the boxes as it was performed here.*

  We tested before, using slightly different models, whether including more predictive variables for the wind improves our model results. We found that doing this decreased the deviance but did not improve the standard error of the rates. However, we checked this again using our current models and found that including the meridional wind instead of only including the zonal wind did improve the standard error. Therefore, we revised our manuscript and now include a GAM TrNcW which includes both zonal and meridional wind instead of TrNcZw which included only zonal wind. We also checked whether including two boxes of proxies (as Dangendorf et al. (2014a) did) improved the standard error compared to taking the difference between these boxes. These

two methods result in similar standard errors. Therefore, we choose to use the method where we take the difference as the model is simpler and the difference between the boxes represents the zonal wind by geostrophy. We thank the author for this useful comment as it helped us improve the manuscript a lot.

- *L195-196: Why are your results in contraction with the ones of Dangendorf et al (2014a) Do you mean that they get a negative trend while you have a positive one in Figure 3b? Are you then comparing your trends from 1836-1928 and 1929-2020 with the ones from 1953-2003 and 1900-2011 from Dangendorf et. al (2014a)? If that is the case, it's not very consistent to compare trends of such different periods. Then the difference in the results would not be only the results of an update in the atmospheric reanalysis, but also from the fact that you are comparing different periods.*

  Indeed, for Figure 3b we find a positive trend due to atmospheric drivers over the whole period. Dangendorf finds a positive trend for the period 1953–2003 (Figure 2c of Dangendorf et al. (2014b)) but a negative trend over the period 1900–2011 (Figure 12 of Dangendorf et al. (2014b)). We will improve our writing such that our message becomes clearer. Whereas we do compare different periods, Figure 3b shows the evolution of the trend over time, an increase in the wind influence on sea level over the full period. We would thus find a positive trend over any period, whereas Dangendorf et al. (2014b) finds a negative trend in Figure 12. We will add the time averages of our results over the same periods to the text.

- *Section 4.3: Was a bit confusing how you start with Figure 4, discusses Table 2, and then moves back to Figure 4. I would suggest have it clearly separated, to be less confusing for the reader (and maybe moving the paragraph starting in L231 to before L215, to introduce the acceleration in the 1960s).*

  Thanks for the comment. We agree that switching between the figure and tables can be confusing. However, the section does make sense in terms of the storyline. We decide to prioritize the storyline here.

- *L210-212: Would it be possible to give a number when you talk about lower/higher uncertainties here? Maybe the average width of the confidence interval for the time series?*

  Indeed, we will improve our text by discussing the time average of the standard error for the different models.

- *L212: Refer here to Figure 4f.*

  The comment will be implemented.

- *L222-227: You talk here in percentages, and table 2 is in probability. Just a suggestion, but maybe you should have Table 2 in % as well, to be consistent.*

  Thanks for the suggestion. We will, however, keep the probabilities in the table and percentages in the text. Our reasons are that it is common practice to show probabilities. However, in the text, we use percentages to explain the meaning of the numbers to people who are not used to p-values.

- *L228-230: This is a very important conclusion of your results. You should emphasize this in the conclusion (sorry if you have done it and I missed it).*

This is emphasized in the last paragraph of the conclusions.

- *L231: typo: 1960".*

Thanks, it will be changed.

- *Figure 3 caption: would be good to add to the caption the line colors as well: "(TrNcZw, orange line) ... (TrNcPd, blue line)". Also, when referring to Figure 1, should be Figure 1b.*

Thanks, it will be changed.

- *Table 2 caption: I guess this is more of a style choice, but I would say that the explanation that starts with the "For example, ..." should be in the main text, and not in the caption of the table.*

We noticed that this table could be wrongly interpreted by people who are not used to p-values. This is why we prefer to keep this additional text here to also make it clear to people who will not read the text thoroughly.

- *L241: "Calafat and Chambers (2013) and Steffelbauer et al (2022).*

Thanks, it will be changed.

- *L246: "results in Calafat and Chambers (2013) and Steffelbauer et al (2022)." not between parenthesis*

Thanks, it will be changed.

- *L259-262 This is a very long sentence (4 sentences) ... Also, 1 paragraph sentences should be avoided.*

We will rewrite this sentence for clarity. However, the discussion is structured to start a new paragraph for a new subject which makes this a rather small paragraph.

- *L267: Maybe also good to refer to the work from Dangendorf et al (2021) here?*

Thank you for this suggestion, we will include the reference.

- *L275: Is the 1.5 the rate of the acceleration period (1975-2000)? That is not shown in Figure 4c. Here also would benefit from having a table with the trends for the different periods (as suggesting in General Comment 2). Also the reason why you divided by 25 (I assume is the number of years from 1975-2000) was not super clear, it would be good to have this better explained, as this is a very important application of the results.*

We will explain how we obtain the rate of 1.5 and why we divide by 25 years. We obtain the acceleration by obtaining the difference in rate (1.5) over a period of 25 years and divide by the period (25).

- *L285-286: This sentence implies that you did test the model using more than one predictive variable for the wind (which was unclear before). If you did test it, I would suggest having these results in the supplementary information, so that the reader can actually see it.*

Good suggestion. The comparison of the different methods is now discussed in the first section

"Comparison of Different GAMs" of the "Results" section.

---

## Author Comment (AC4)

**DOI:** 10.5194/egusphere-2022-935
**Version:** March 3, 2023
**Title:** The acceleration of sea-level rise along the coast of the Netherlands started in the 1960s
**Authors:** Iris Keizer, D. Le Bars, C. de Valk, A. Jüling, R. van de Wal, and S. Drijfhout

**Point by point reply to reviewer #2**

We thank the reviewer for their careful reading and beneficial comments on the manuscript and address their points below. Since we added a table to the revised manuscript (Table 2), Table 2 of the old version is Table 3 in the revised version. Also, two appendix sections are added (Appendix A and Appendix B), making Appendix A of the old version Appendix C of the revised version. Some major changes have been made to the manuscript. We obtained the most recent available tide gauge records and atmospheric data. Therefore, the time series are extended to 2021 and, respectively, 2022. We checked whether using more predictive variables in our models including wind as a predictive variable improved the results. We found that adding the meridional wind to the model that only included the zonal wind improved the results. Therefore, we now include a local wind model that includes both the zonal and meridional wind. We included results on the nodal tide influence in the appendix as well as the sea-level rates for the individual tide gauge stations.

**Summary**

*The paper estimates the sea-level rise (SLR) using the average of six tide gauges along the coast of the Netherlands, from 1890 to 2020. For this purpose, the authors used four Generalized Additive Models – GAMs, to estimate the sea-level trend, as well as the influence of the lunar nodal cycle and zonal wind on sea level, the latter using two different approaches. Results indicate an acceleration of SLR starting in the 1960's, which protrudes once the tidal and wind effects on sea level are removed. Besides, they show that wind effects force a long-term SLR, as well as a low- frequency variability modulating sea level by about 1 cm, which is related to Sea Surface Temperature variations in the North Atlantic. I find the aim of the paper very relevant, as the assessment of coastal rates of SLR and its acceleration due to anthropogenic forcing, is essential to obtain adequate projections of future sea levels, necessary to perform effective coastal adaptation strategies. Besides, I find the paper scientifically relevant, especially for the use of GAMs to estimate the sea-level trends, allowing them to isolate the influence of particular variables. The paper is well written and has an excellent presentation quality. However, there are few comments I would like the authors to consider before submitting this preprint for publication in a scientific journal.*

**Main Comments**

- *L103. Through the entire paper, the assessment of the sea-level along the coast of the Netherlands uses the average of the six tide gauges. I suggest to include a comment about the reasons of using this average instead of using the individual time series (or both). Furthermore, I think it would be interesting to assess if the SLR trends and acceleration observed in the averaged time series stand, if the same method is applied to the individual tide gauge time series. In particular, the wind effect on sea level could be very different from one station to the other, depending not only in spatial differences in the wind forcing, but also due to coastline configuration, bathymetric differences, among others.*

Thank you for this useful comment. In the paper, we will expand on our reasoning to use the average of the six tide gauges. The aim of this paper is to study the sea-level rate for the Netherlands to better understand sea-level projections and improve adaptation choices. The sea-level rates are useful for policymakers, and our results are already used for sand suppletion. Indeed, the rates could be different for the individual stations, but it does not fit the scope of the current study to study those differences. We aim to keep the paper concise and looking at the sea-level rates of the individual stations adds complexity because of vertical land motion and small-scale ocean processes. Using the average of the six stations is sufficient to study the acceleration of the sea-level rates. Moreover, since CMIP6 climate models don't have a resolution on the local scale there is no use to study the individual stations to relate the sea-level rates studied in our work to the projections. We will, however, also include the rates of sea level for the individual stations in the appendix.

- *L112. Two sinusoidal waves in opposition of phase with the 18.6 year period were used to assess the nodal effect on sea-level. Although in L120 the authors recognize this approach might remove some additional variability around the nodal period, I suggest the authors to include in the results section, at least one paragraph describing the amplitude and phase lag of the nodal cycle clearly seen in the TrNc GAM. This result should be compared to what is known about this cycle in literature.* (l.114) Thank you for this useful comment. We will include a section in the appendix on our results for the nodal cycle amplitude and phase lag and a comparison to the equilibrium tide, as suggested. We will add a figure showing the statistical estimation of the nodal tide resulting from our models TrNcZw and TrNcPd and the equilibrium tide to the section in the appendix.

- *Appendix A. In L321 authors indicate they assess the low-frequency relation between wind influence on sea level and low-frequency SST. At this point I think authors should guide the reader, indicating the physical relation between these two variables they are trying to expose. Later in L345, authors offer a physical explanation for the relation found between SST in the north box and zonal wind. In my view, the hypothesis that changes in the meridional temperature gradient strength the jet stream, is not strongly presented. Due to air-sea interaction, SST has an inverse relationship with atmospheric pressure at sea level. As used in the paper (TrNcPd), changes in the meridional atmospheric pressure gradient modulate zonal wind in the region. Therefore, the relation found between wind influence on sea level and SST is probably possible due to variations in the atmospheric pressure gradient, what in the region is measured by NAO. This reasoning also supports the stronger correlations found in TrNcPd when compared to TrNcZw. I suggest authors to review literature about the relation between NAO and SST (air-sea coupling) in the North Sea (e.g. doi/10.1029/2022JD037270), to present a stronger case in the Appendix. Changes in the Appendix might force some changes in the main paper.*

We agree that this appendix needs some clarifications and we will rewrite it to elaborate on the NAO-SST-wind relationship and its bearing on the multidecadal wind-influence-on-sea-level signal that we show in Fig. 3d. We will explain the two relevant, well-established modes of variability, NAO and AMV, in more detail and explain our analysis results better. The pressure difference component of our GAM *TrNcPd* is closely related to the NAO which also varies at multidecadal time scales. Therefore, we will add some more discussion on the low-frequency behaviour of the NAO (e.g. doi:10.1007/s00382-014-2237-y). As for the SST influence on the wind, we will sketch how changing North Atlantic SST (pattern)s can influence the Dutch sea level via winds as you suggest. This section is currently not finished, but we will supply a new draft within a week.

**Specific Comments**

- *L4. ...covering the period 1890-2020.*

  Thanks, we will change this.

- *L13. Verify a typing error.*

  Thanks, we will change this.

- *L134. Authors assess sea level from 1890 to 2020. Suddenly in this line, they mention that wind effects on sea level are assessed from 1836 to 2020. I suggest authors to include a comment about the reasons of extending in time the assessment of the wind forcing.*

  Thanks, we will clarify that once the regression coefficients are obtained, we can obtain the wind influence on sea level over the entire time series of the wind data, which covers 1836 - 2021. However, in the method section, we only focus on the fact that both atmospheric reanalysis datasets are combined using a linear bias correction method. In the paragraph 'analysis of model output', we will clarify how we obtain the wind influence on sea level.

- *L189. In this line authors assess the wind-driven trend in sea level for the 1928-2020 period. However, Figure 3b show trends for two periods. I suggest authors to include a comment about the wind-driven sea-level trends observed during the first period. I think this is important especially due to the large difference in the trends observed in TrNcPd.*

  Indeed, we will include a comment on the trends for the first period.

- *L232. The authors speculate about the reasons behind the SLR decreasing rate observed from 1900 to 1960. Please consider to move this explanation to the discussion section.*

  We will move this paragraph to the discussion section.

- *Table 2 legend. The probability in the third line is 0.23.*

  Thanks, we will change this.

- *L246. Verify the referencing.*

  We corrected this in the text.

- *L274. 2000 onwards, ...*

  We will rephrase the sentence for clarity.

- *Figure A1 legend. Define the AMV acronym, indicating that the area is shown with the black limits. Try to use colors that can be easily distinguished in panels' b and d.*

  Thanks. We will clarify the appendix by adding the AMV acronym description, the area indication, and we will improve the colours in the figure.

---

## Referee Report (RR1)

Review of the manuscript: "**The acceleration of sea-level rise along the coast of the Netherlands started in the 1960s**" by Iris Keizer et al.

This is my second review of the paper. I want to thank the authors' work and effort answering my recommendations to the first version of the manuscript, which I believe helped to improve the paper. Authors answered to my main comments. Frist, they included an Appendix with the tide gauge's individual SLR rates, giving a clear explanation of why they decided to use the six tide gauges average through the paper. In addition, at the end of section five, they discuss the differences in the SLR rates obtained from the individual tide gauges when compared to their average, giving the reader the complete information to ponder the paper's results. Second, authors expanded the information related to the nodal effect on sea-level in an Appendix, what I believe also helps to explain some paper's details which might be important to some particular readers. Finally, Appendix C in my view presents a stronger case of the possible drivers of low frequency wind-driven sea level variations in the Netherland's coasts, found in this research. Therefore, I have no main comments on the new version of the paper. However, I present some specific comments and minor recommendations to the authors for them to consider include into the final version of the paper.

Specific comments and minor recommendations:

- L50. Appendixes are named as "Appendix" and as "App." (L123). Please use the same naming. Besides, verify Appendixes are organized in the same order as they are mentioned in the main text.

- L56. …meridional surface wind velocities …

- L109. Tables are named as "Tbl." and as "Table" (L149). Please use the same naming for tables and figures.

- L127. … nodal tide and wind effects.

- Table 1 legend. Third line. "… predictive variables for wind effects.". Use the same lower case Phi symbol, as in the formulas.

- Figure 2 legend. First line. "… with three sea level time series obtained from the Generalised …".

- L220. … (1988) with …

- Figure 3 legend. Last row. (d) Detrended and smoothed time series shown in (a).

- Table 2. Please mention this table in the results section. Table 2 is only mentioned in the Conclusions section of the main text (L333).

- L258. Calafat and Chambers (2013) and Steffelbauer et al. (2002).

- L298. In this version is (App. A).

- L330. Fig. 4.

- L333-L335. Consider moving this sentence to the Discussion section as this is not a conclusion from the paper.

- Appendix A. I find the difference between the nodal tide from the GAMs models and the equilibrium tide too large. For clarity, please mention or give a reference to the method used to obtain the equilibrium tide from the individual tide gauge records. In my view, to accurately assess the equilibrium tide, the nodal cycle has to be calculated in each of the most important lunar tidal constituents (e.g. https://doi.org/10.1016/j.csr.2009.10.006). To assess the nodal effect on sea level, these constituents' nodal cycles (amplitude and phase) should be added. I understand that such calculations are probably out of the scope of the paper. However, a reference to the method used to compute the equilibrium tide I think is needed as in my view, the nodal cycle obtained from the GAMs somehow is a novel method to assess the nodal cycle effect in long annual sea level records.

Please verify colors in Fig. A1(a) and its legend. Besides, I found legend in Fig. A1(b) confusing. Consider a different legend (e.g. OSL-TrW; OSL-Nt-TrW; OSL-ET-TrW; OSL-Et; OSL for observed sea level). The last spectra is not mentioned in the legend.

In the appendix text the spectra from "Et" is not mentioned but it is shown in Fig A1(b). I do not understand why the "Et" residual spectra has low energy in the nodal period, while the "TrEtW" residual spectra has much more energy in this period.

L356 – 357. Please rephrase the sentence. I understand that the TrNtW and TrEtW models use the trend and wind as predictive variables, but they differ because the former uses the nodal tide obtained from the GAMs, while the latter uses the equilibrium tide. However, this is not what the sentence indicates.

-Appendix B. Consider including in Figure B1, the SLR obtained from the six tide gauge average using the TrNtW model, as shown in Figure 4e and f (orange line). This would be useful to the reader to observe individual tide gauge sea level rate differences form the average. In Figure B1 legend, second line, replace TrNtZw for TrNtW.

---

## Author Response (AR2)

**DOI:** 10.5194/egusphere-2022-935
**Version:** June 2, 2023
**Title:** The acceleration of sea-level rise along the coast of the Netherlands started in the 1960s
**Authors:** Iris Keizer, D. Le Bars, C. de Valk, A. Jüling, R. van de Wal, and S. Drijfhout

**Point by point reply to reviewer #2**

We Thank the reviewer for their additional comments on the manuscript and address these points below. All comments have been implemented and contribute to the quality of the manuscript.

**Summary**

*This is my second review of the paper. I want to Thank the authors' work and effort answering my recommendations to the first version of the manuscript, which I believe helped to improve the paper. Authors answered to my main comments. Frist, they included an Appendix with the tide gauge's individual SLR rates, giving a clear explanation of why they decided to use the six tide gauges average through the paper. In addition, at the end of section five, they discuss the differences in the SLR rates obtained from the individual tide gauges when compared to their average, giving the reader the complete information to ponder the paper's results. Second, authors expanded the information related to the nodal effect on sea-level in an Appendix, what I believe also helps to explain some paper's details which might be important to some particular readers. Finally, Appendix C in my view presents a stronger case of the possible drivers of low frequency wind-driven sea level variations in the Netherland's coasts, found in this research. Therefore, I have no main comments on the new version of the paper. However, I present some specific comments and minor recommendations to the authors for them to consider include into the final version of the paper.*

**Specific Comments and Minor Recommendations**

- *L50. Appendixes are named as "Appendix" and as "App." (L123). Please use the same naming. Besides, verify Appendixes are organized in the same order as they are mentioned in the main text.*

  Thank you, we implemented this suggestion.

- *L56. ...meridional surface wind velocities ...*

  Thank you, we implemented this suggestion.

- *L109. Tables are named as "Tbl." and as "Table" (L149). Please use the same naming for tables and figures.*

  Thank you, we implemented this suggestion.

- *L127. . . . nodal tide and wind effects.*

  Thank you, we implemented this suggestion.

- *Table 1 legend. Third line. ". . . predictive variables for wind effects.". Use the same lower case Phi symbol, as in the formulas.*

  Thank you, we implemented this suggestion.

- *Figure 2 legend. First line. ". . . with three sea level time series obtained from the Generalised . . .".*

  Thank you, we implemented this suggestion.

- *L220. . . . (1988) with . . .*

  Thank you, we implemented this suggestion.

- *Figure 3 legend. Last row. (d) Detrended and smoothed time series shown in (a).*

  Thank you, we implemented this suggestion.

- *Table 2. Please mention this table in the results section. Table 2 is only mentioned in the Conclusions section of the main text (L333).*

  In Section 4.3 on Rates of Sea Level Rise, we have added references to Table 2. Firstly, in the first sentence (line 227), where we mention that the rates of sea level rise obtained from each of the four models are shown in Fig. 4, and the average rates over different periods are provided in Table 2. Secondly, in line 239, we present the specific rates of sea level rise for the model TrNtW over different periods, accompanied by references to both Fig. 4 and Table 2.

- *L258. Calafat and Chambers (2013) and Steffelbauer et al. (2002).*

  Thank you, we implemented this suggestion.

- *L298. In this version is (App. A).*

  Thank you, we implemented this suggestion.

- *L330. Fig. 4.*

  Thank you, we implemented this suggestion.

- *L333-L335. Consider moving this sentence to the Discussion section as this is not a conclusion from the paper.*

  We moved the sentence to the discussion section. As it didn't fit into any of the existing paragraphs, it is added as a new paragraph with some context.

- *Appendix A. I find the difference between the nodal tide from the GAMs models and the equilibrium tide too large. For clarity, please mention or give a reference to the method used to obtain the equilibrium tide from the individual tide gauge records. In my view, to accurately assess the equilibrium tide, the nodal cycle has to be calculated in each of the most important lunar tidal constituents (e.g. https://doi.org/10.1016/j.csr.2009.10.006). To assess the nodal effect on sea level, these constituents' nodal cycles (amplitude and phase) should be added. I understand that such calculations are probably out of the scope of the paper. However, a reference to the method used to compute the equilibrium tide I think is needed as in my view, the nodal cycle obtained from the GAMs somehow is a novel method to assess the nodal cycle effect in long annual sea level records.*

  We have addressed your valuable comment by incorporating the reference for the estimation of the equilibrium tide to Section Appendix A: Nodal Effects on Sea Level. The equilibrium tide computation in our study follows the approach outlined in Woodworth, 2012 (https://doi.org/10.2112/JCOASTRES-D-11A-00023.1). This method was also utilized in the budget calculations conducted by Frederikse et al. (https://doi.org/10.1002/2016GL070750).

- *Please verify colors in Fig. A1(a) and its legend. Besides, I found legend in Fig. A1(b) confusing. Consider a different legend (e.g. OSL-TrW; OSL-Nt-TrW; OSL-ET-TrW; OSL-Et; OSL for observed sea level). The last spectra is not mentioned in the legend.*

  After reviewing the colors and legend of Figure A1, we have made adjustments to enhance clarity. Specifically, we have modified the appearance of the TrNtPd line by making it a broken line, thereby improving the visibility of the plot. Furthermore, we have updated the legend of A1(b) to clearly indicate when the residual is plotted and which signal is represented. Additionally, we have included a statement in the caption of Figure A1 mentioning that the spectrum of the equilibrium tide (Et) is plotted.

- *In the appendix text the spectra from "Et" is not mentioned but it is shown in Fig A1(b). I do not understand why the "Et" residual spectra has low energy in the nodal period, while the "TrEtW" residual spectra has much more energy in this period.*

To provide a clearer explanation, we have refined the text of Appendix A by incorporating the following lines: "Due to the use of the multitaper method, the spectrum of *Et* does not manifest as a single peak at 18.613 years; rather, it appears as a broad peak centered around that period due to the windowing effect. As anticipated, in the vicinity of the equilibrium tide period, the residuals of *TrW* retain the majority of the energy since this model does not encompass the nodal tide, thereby leaving the nodal tide signal in the residuals. Conversely, when the equilibrium tide is incorporated into the model (*TrEtW*), the nodal tide signal should no longer persist in the residuals, resulting in a reduced power in the spectrum around the equilibrium tide period. We observe that indeed less power remains around this period, and the power removed is equivalent to the power of the equilibrium tide (*Et*). However, a substantial amount of energy remains compared to the utilization of *TrNtW*. This outcome underscores our decision to employ a statistical estimation for the nodal tide."

- *L356 – 357. Please rephrase the sentence. I understand that the TrNtW and TrEtW models use the trend and wind as predictive variables, but they differ because the former uses the nodal tide obtained from the GAMs, while the latter uses the equilibrium tide. However, this is not what the sentence indicates.*

  We have added an explanatory sentence to clarify the context: "The reconstructed sea level is obtained for three models: TrW, which includes the sea-level trend and zonal and meridional wind but excludes the nodal tide; TrNtW, which includes the sea-level trend, zonal and meridional wind, and the nodal tide estimated using a particular method; and TrEtW, which employs the same predictive variables as TrW but corrects the sea level data using the equilibrium tide. By comparing the results of TrEtW and TrNtW, which differ only in their method of obtaining the nodal tide, we can examine the impact of the chosen method on the resulting nodal tide estimation."

- *Appendix B. Consider including in Figure B1, the SLR obtained from the six tide gauge average using the TrNtW model, as shown in Figure 4e and f (orange line). This would be useful to the reader to observe individual tide gauge sea level rate differences form the average. In Figure B1 legend, second line, replace TrNtZw for TrNtW.*

  Thank you, excellent suggestion, we included the average to the plots.